# Self-contradictory Hallucinations of LLMs: Evaluation, Detection and Mitigation

**Niels Mündler, Jingxuan He, Slobodan Jenko & Martin Vechev**
Department of Computer Science, ETH Zurich, Switzerland
`nmuendler@student.ethz.ch, jingxuan.he@inf.ethz.ch,`
`sjenko@student.ethz.ch, martin.vechev@inf.ethz.ch`

## Abstract

Large language models (large LMs) are susceptible to producing text that contains hallucinated content. An important instance of this problem is self-contradiction, where the LM generates two contradictory sentences within the same context. In this work, we present a comprehensive investigation into self-contradiction for various instruction-tuned LMs, covering evaluation, detection, and mitigation. Our primary evaluation task is open-domain text generation, but we also demonstrate the applicability of our approach to shorter question answering. Our analysis reveals the prevalence of self-contradictions, e.g., in 17.7% of all sentences produced by ChatGPT. We then propose a novel prompting-based framework designed to effectively detect and mitigate self-contradictions. Our detector achieves high accuracy, e.g., around 80% F1 score when prompting ChatGPT. The mitigation algorithm iteratively refines the generated text to remove contradictory information while preserving text fluency and informativeness. Importantly, our entire framework is applicable to black-box LMs and does not require retrieval of external knowledge. Rather, our method complements retrieval-based methods, as a large portion of self-contradictions (e.g., 35.2% for ChatGPT) cannot be verified using online text. Our approach is practically effective and has been released as a push-button tool to benefit the public at `https://chatprotect.ai/`.

## 1 Introduction

Large language models (large LMs) are pretrained on massive text corpora (Chowdhery et al., 2022; Zhang et al., 2022; Touvron et al., 2023a) and fine-tuned to follow human instructions (Ouyang et al., 2022). Instruction-tuned LMs, such as ChatGPT (OpenAI, 2023a), have demonstrated remarkable zero-shot capabilities in solving natural language tasks (Bang et al., 2023; Qin et al., 2023; Zhong et al., 2023; OpenAI, 2023b). Therefore, they are being increasingly integrated into various aspects of daily life, including online search and professional environments (Mehdi, 2023; NerdyNav, 2023; Spataro, 2023). However, even widely adopted LMs such as ChatGPT and GPT-4 are prone to generating nonsensical or unfaithful content, commonly referred to as hallucinations (Bang et al., 2023; OpenAI, 2023b; Borji, 2023). This raises significant concerns regarding the trustworthiness of LMs (Weidinger et al., 2021; Zhuo et al., 2023; Reuters, 2023; Bose & Shepardson, 2023). Detecting and mitigating hallucinations still remains an open challenge (Ji et al., 2023), especially for state-of-the-art, proprietary LMs accessible only as black boxes (OpenAI, 2023a;b; Anthropic, 2023).

**Reasoning about Self-contradictory Hallucinations** This work focuses on an important type of hallucination called self-contradiction, which occurs when an LM generates two logically inconsistent sentences given the same context. Our key insight is that self-contradiction can be leveraged to conveniently tackle non-factual model outputs. We exploit the fact that self-contradiction is guaranteed to reveal non-factuality, because two contradicting sentences cannot be simultaneously correct (Dowden, 2011). Moreover, removing conflicting information from a pair of contradictory sentences strictly decreases non-factuality. Importantly, both contradiction detection and removal can be accomplished by performing logical reasoning, a particular strength of modern LMs (Liu et al., 2023). This frees us from relying on externally retrieved knowledge (Peng et al., 2023) or comparing tens of samples (Manakul et al., 2023b), which can be difficult or too expensive in practice.

Table 1: Two self-contradictory hallucinations generated by ChatGPT. Sentences marked with green color (resp., red color) are factually correct (resp., wrong). Our framework successfully triggers, detects, and mitigates both hallucinations. We provide additional, longer examples in Appendix C.

| | |
|---|---|
| Prefix | The PlayStation 4 (PS4) is a home video game console developed by Sony. |
| Sentence pair | Released in 2013, it is the eighth generation of consoles in the PlayStation series. |
| | It is the fourth generation of the PlayStation console series. |
| Mitigation | The PlayStation 4 (PS4) was released in 2013. |
| Prefix | Gwen Jorgensen is a retired professional triathlete from the United States. |
| Sentence pair | She currently lives in Minnesota with her husband, Patrick Lemieux, and their children. |
| | She currently lives in Portland, Oregon with her husband and two children. |
| Mitigation | She currently lives with her husband and children. |

Based on the above insight, we propose a three-step pipeline for reasoning about self-contradictions. To trigger self-contradictions, we employ appropriate constraints to generate relevant sentence pairs. Then, we explore various existing prompting strategies to detect self-contradictions (Wei et al., 2022; Kojima et al., 2022; Wang et al., 2023). Lastly, we develop an iterative mitigation procedure that makes local text edits to remove contradictory information, while preserving other important text qualities such as fluency and informativeness. Since our framework operates through prompting, it is directly applicable to state-of-the-art black-box LMs.

**Significance of Self-contradiction** We perform an extensive evaluation on four modern LMs: GPT-4 (OpenAI, 2023b), ChatGPT (OpenAI, 2023a), Llama2-70B-Chat (Touvron et al., 2023b), and Vicuna-13B (Chiang et al., 2023). We primarily focus on open-domain text generation, a task that involves producing long text using LMs' internal knowledge (Petroni et al., 2019; Cohen et al., 2023a), where trustworthiness is highly desired but challenging to achieve (Ji et al., 2023). Our evaluation highlights the significance of self-contradiction. Specifically, self-contradictions are prevalent across the evaluated LMs, e.g., in 17.7% of all sentences generated by ChatGPT in open-domain text generation. A large portion of these (e.g., 35.2% for ChatGPT) cannot be verified using Wikipedia or text obtained by web search. In other words, our work serves as a valuable complement to retrieval-based approaches (Shuster et al., 2021; Peng et al., 2023; Min et al., 2023).

**Effective Detection and Mitigation** Our proposed framework is highly effective at detecting (e.g., ∼80% F1 score) and mitigating self-contradictions (e.g., reducing up to of 89.5% self-contradictions while maintaining text informativeness and fluency). In Table 1, we present two instances of ChatGPT-generated self-contradictions that are successfully triggered, detected, and mitigated by our method. In Appendix C, we provide various longer examples, at both sentence and text levels.

**Generality** We further apply our method to question answering (Mallen et al., 2023). The results show that our approach accurately (e.g., 74.2% to 83.8% precision) detects a significant number (e.g., 12.7% to 38.0%) of self-contradictions, for both vanilla and retrieval-augmented question answering.

**User-friendly Tool and Open-source Repository** The effectiveness and generality of our framework underscore its practicality. We release a user-friendly tool that produces warnings for hallucinations and automatically mitigates them, accessible at https://chatprotect.ai/. Our code and datasets are publicly available on GitHub at https://github.com/eth-sri/ChatProtect.

## 2 RELATED WORK

In this section, we discuss works that are closely related to ours.

**Large Language Models** Training modern LMs requires two steps: pretraining (Chowdhery et al., 2022; Zhang et al., 2022; Touvron et al., 2023a) and instruction-tuning (Ouyang et al., 2022). Commercial instruction-tuned LMs (OpenAI, 2023a;b; Anthropic, 2023) are typically proprietary and only provide black-box accesses. Starting from Alpaca (Taori et al., 2023), open-source instruction-tuned LMs have emerged, such as Vicuna (Chiang et al., 2023) and Llama2 (Touvron et al., 2023b). Instruction-tuned LMs are increasingly used in daily life (Mehdi, 2023; NerdyNav, 2023; Spataro, 2023), because they are powerful in text reasoning (Bang et al., 2023; Qin et al., 2023; Zhong et al.,

2023; Liu et al., 2023). Comprehensive benchmarks, such as MMLU (Hendrycks et al., 2021) and HELM (Liang et al., 2022), have been proposed to thoroughly assess LMs' overall capabilities. Our work extends beyond these by focusing specifically on LMs' power to address hallucinations.

**Hallucinations in Natural Language Generation** According to the survey by Ji et al. (2023), hallucination is a common issue in different natural language generation tasks, including translation (Zhou et al., 2021), summarization (Maynez et al., 2020; Kryscinski et al., 2020), dialogue (Dinan et al., 2019; Shuster et al., 2021), and question answering (Lin et al., 2022; Kuhn et al., 2023; Cohen et al., 2023b; Mallen et al., 2023). Two other works have explored hallucinations in open-domain text generation (Lee et al., 2022; Manakul et al., 2023b). While Lee et al. (2022) require access to internals of training and inference, our work is designed to be applicable to black-box LMs. Compared with Manakul et al. (2023b), our framework is significantly more accurate in hallucination detection (as shown in Appendix B) and additionally addresses mitigation.

## 3 DEFINING AND MOTIVATING SELF-CONTRADICTIONS

This section defines self-contradiction of LMs and presents our motivations for addressing them.

**Language Models** We consider a language model (LM) that generates text consisting of a sequence of sentences: $\mathbf{x} = [x_1, x_2, \ldots, x_{|\mathbf{x}|}]$. Typically, the text generation process is conditioned on a user prompt $p$ that specifies the task to accomplish. For the generation of multiple sentences $\mathbf{x}$, we denote this process as $\mathbf{x} \sim \mathrm{LM}(\cdot \mid p)$. Similarly, for a single sentence $x$, we use $x \sim \mathrm{LM}(\cdot \mid p)$.

**Self-contradiction of LMs** We examine a pair of sentences $(x, x')$ generated by the LM. For $x$ and $x'$ to be contradictory, they must describe the same subject. Therefore, we constrain their generation by providing a context $c$ in the form of a prompt. That is, $x \sim \mathrm{LM}(\cdot \mid c)$ and $x' \sim \mathrm{LM}(\cdot \mid c)$. We define $(x, x')$ to be a self-contradiction of the LM when the two sentences are logically inconsistent. Our definition requires both sentences to be generated from the same LM, in contrast to prior works that use external sentences (i.e., those not generated by the LM) for hallucination evaluation (Elazar et al., 2021; Azaria & Mitchell, 2023). As opposed to self-consistency over different chain-of-thought reasoning paths (Wang et al., 2023), our definition captures factual statements in broader contexts.

**Self-contradiction vs. Non-factuality** When $x$ and $x'$ contradict each other, at least one of them is guaranteed to be factually incorrect (Dowden, 2011). Based on our annotations, in 63.1% of the self-contradictions generated by ChatGPT, even both sentences are non-factual. Since both $x$ and $x'$ originate from the same LM, their contradiction must expose the LM's non-factuality. Exploiting this insight a step further, removing conflicting information from a pair of contradictory sentences strictly decreases non-factuality. These key properties empower us to detect and mitigate non-factuality via self-contradiction, using only two sampled sentences each time.

Like ours, related works leverage sampling-based methods to detect non-factuality (Kuhn et al., 2023; Manakul et al., 2023b; Cohen et al., 2023b). However, they do not benefit from the insights discussed above regarding the connection between non-factuality and self-contradiction. As a result, they require tens of samples to achieve reasonable detection accuracy and cannot handle mitigation.

**Self-contradiction vs. Knowledge Retrieval** One common approach for addressing hallucinations depends on the retrieval of external knowledge to identify non-factual content or guide factual text generation (Dinan et al., 2019; Shuster et al., 2021; Peng et al., 2023; Min et al., 2023; Mallen et al., 2023). However, effective knowledge retrieval is challenging and costly in practice (Zhou et al., 2022; Ji et al., 2023; Mallen et al., 2023). Meanwhile, the detection and mitigation of self-contradictions can be achieved solely through logical reasoning, a particular strength of the latest LMs even in a zero-shot manner (Liu et al., 2023). Indeed, our evaluation shows that a significant portion of real-world self-contradictions, e.g., 35.2% for ChatGPT, cannot be verified or refuted using relevant online text. Moreover, as demonstrated in Section 6.3, our approach is able to detect a large number of self-contradictions even in retrieval-augmented generations (Mallen et al., 2023). This makes our work a valuable complement to retrieval-based approaches.

## 4    TRIGGERING, DETECTING AND MITIGATING SELF-CONTRADICTIONS

We consider two LMs, gLM, which generates text $\mathbf{x} = [x_1, x_2, \ldots, x_{|\mathbf{x}|}]$, and aLM, an analyzer LM. In this section, we describe our algorithms for triggering self-contradictions of gLM, as well as for detecting and mitigating them using aLM. From the algorithms, we decouple four important utility functions that involve solving different text reasoning tasks. Below, we provide generic definitions of these utility functions. In Section 5, we discuss their concrete implementations.

- `extract_contexts`$(x_i, \mathbf{x})$: extracts a list of contexts for sentence $x_i$, possibly using metadata information from $\mathbf{x}$, such as the entity of $\mathbf{x}$.
- `gLM.gen_sentence`$(c)$: queries gLM to generate a new sentence $x'_i$ compatible with context $c$.
- `aLM.detect`$(x_i, x'_i, c)$: invokes aLM to predict if $x_i$ and $x'_i$ are contradictory within context $c$.
- `aLM.revise`$(x_i, x'_i, c)$: receives a pair of contradictory sentences $(x_i, x'_i)$ within context $c$. The method invokes aLM to generate a revised version of $x_i$ where the conflicting information between $x_i$ and $x'_i$ is removed. The revised sentence is also expected to retain as much non-conflicting information as possible and to be coherent with context $c$.

**Trigger**    In typical usages, gLM generates each sentence only once. To trigger self-contradictions, we need to appropriately query gLM to generate a second sentence to form a pair. We develop Algorithm 1 to achieve this purpose. At Line 2, it iterates over the sentences of $\mathbf{x}$. For each sentence $x_i$, it calls `extract_contexts` to obtain a list of contexts (Line 3). Then, at Line 4, it runs `gLM.gen_sentence` that returns an alternative sentence $x'_i$ that aligns with context $c$. At Line 5, the resulting sentence pair $(x_i, x'_i)$ and its context $c$ are yielded following Python semantics. The context $c$ plays a crucial role in the generation of $x'_i$. Our goal is for $c$ to effectively constrain $x'_i$ to have the same scope as $x_i$. Meanwhile, we desire $c$ to provide an appropriate level of freedom, allowing gLM to generate $x'_i$ that contradicts $x_i$. We elaborate on achieving this dual objective in Section 5.

**Detection**    Detecting self-contradictions is done by calling `aLM.detect` on the output of Algorithm 1. This procedure is closely related to natural language inference (NLI) (Bowman et al., 2015; Dagan et al., 2005), which involves predicting the relationship between a premise and a hypothesis: entailment, neutrality, or contradiction. Our detection differs from general NLI: (i) we require that both sentences are generated from the same LM; (ii) we consider the context in which the two sentences appear during detection; (iii) our detection output is binary: contradictory or not.

**Mitigation**    Our mitigation approach performs local revisions of all sentences in $\mathbf{x}$ using the process shown in Algorithm 2 to remove self-contradictory information. If `aLM.detect` predicts a self-contradiction, we utilize `aLM.revise` to eliminate the contradictory information from the sentence pair. We do not change sentences without detected self-contradictions to maintain other important text qualities of $\mathbf{x}$, such as fluency and informativeness.

To further minimize self-contradictions, we repeat the above step on the updated version of $\mathbf{x}$, creating an iterative procedure outlined in Algorithm 3. In each iteration, we invoke `mitigate_one` from Algorithm 2 for all sentence pairs $(x_i, x'_i)$ and contexts $c$ (Line 6). The output of `mitigate_one` is re-assigned to $x_i$ and ultimately written back into the text at $\mathbf{x}$. After a certain number of iterations (e.g., 3 in our experiments), the mitigation process is expected to converge, even though there may be a small number of remaining self-contradictions identified by `aLM.detect`. In such cases, we remove the corresponding sentences. This generally does not compromise the quality of the resulting text, as shown in Table 10 of Appendix B. After this step, all predicted self-contradictions are eliminated.

## 5    INSTANTIATION TO TEXT GENERATION TASKS

Our framework is built on prompting, making it suitable for state-of-the-art and proprietary black-box LMs. In this section, we instantiate these prompts on our primary evaluation task, open-domain text generation (Manakul et al., 2023b; Lee et al., 2022), where we ask LMs to generate long text with dozens of facts for various entities. Focusing on this task offers three key advantages: (i) it closely aligns with the practical use of modern LMs in generating text across diverse domains (Mehdi, 2023; NerdyNav, 2023; Spataro, 2023); (ii) it is well-suited for analyzing self-contradiction as it

**Algorithm 1:** Triggering self-contradictions for text generated by gLM.

```
1  procedure trigger(x)
      input  : x, text generated by gLM.
      output : a sequence of tuples (x_i, x'_i, c).
2     for x_i in x do
3        for c in extract_contexts(x_i, x) do
4           x'_i = gLM.gen_sentence(c)
5           yield (x_i, x'_i, c)
```

**Algorithm 2:** Mitigating self-contradictions for one pair of gLM-generated sentences.

```
1  procedure mitigate_one(x_i, x'_i, c)
      input  : (x_i, x'_i), a pair of gLM-generated
               sentences coherent with context c
      output : a revised version of x_i with
               self-contradictions removed.
2     if aLM.detect(x_i, x'_i, c) then
3        return aLM.revise(x_i, x'_i, c)
4     else
5        return x_i
```

**Algorithm 3:** Iterative mitigation of self-contradictions for text generated by gLM.

```
1  procedure mitigate_iter(x, n)
      input  : x, gLM-generated text.
               n, the number of mitigation iterations.
      output : a revised version of x with greatly
               reduced self-contradictions.
2     repeat n times with y = []
3        for x_i in x do
4           for c in extract_contexts(x_i, x) do
5              x'_i = gLM.gen_sentence(c)
6              x_i = mitigate_one(x_i, x'_i, c)
7           y.append(x_i)
8        x = y
9     return x
```

```
Prompt:
  Here is the start of a description about William T. Freeman:
  William T. Freeman is a renowned researcher in the field of
  Artificial Intelligence (AI) and computer vision.

  Please generate the next sentence of this description.
  The generated sentence must fill the gap in this triple:
  (He; was born; _)

gLM: He was born in 1960.
```

Figure 1: Our prompt for `gLM.gen_sentence`.

```
Prompt:
  Here is the start of a description about William T. Freeman.
  Then follow two statements.

  Description:
  William T. Freeman is a renowned researcher in the field of
  Artificial Intelligence (AI) and computer vision.

  Statement 1:
  He was born on August 15, 1955, in the United States.

  Statement 2:
  He was born in 1960.

  Please explain if the statements about William T. Freeman
  are contradictory. Provide your explanation only.

aLM:
  The statements about are contradictory.
  Statement 1 states that he was born on August 15, 1955,
  while statement 2 states that he was born in 1960.
  These two statements cannot both be true at the same time.

Prompt: Please conclude with Yes or No.

aLM: Yes.
```

Figure 2: Our prompt for `aLM.detect`.

```
Prompt:
  Here is the start of a description about William T. Freeman.
  William T. Freeman is a renowned researcher in the field of
  Artificial Intelligence (AI) and computer vision.

  Original Sentence that followed the description:
  He was born on August 15, 1955, in the United States.

  However, there is a contradiction with this sentence:
  He was born in 1960.

  Remove the conflicting information from the sentence.
  Maintain other information and text fluency.

aLM: William T. Freeman was born in the United States.
```

Figure 3: Our prompt for `aLM.revise`.

requires LMs to untangle their internal knowledge (Petroni et al., 2019; Cohen et al., 2023a); (iii) it is challenging to deal with long text (Ji et al., 2023). At the end of this section, we also discuss the generic instantiation of our approach to other tasks, such as question answering.

**Generating Initial Text**    First, we query gLM to generate text $x$ for user-given prompts. For our open-domain text generation task, the user prompt takes the form "Please tell me about $t$", where $t$ is an entity from a certain domain. After receiving the user prompt, gLM generates a sequence of sentences that reflect its internal knowledge about $t$ in an encyclopedic style.

**Defining Context**    Given the gLM-generated text $x$ and a sentence $x_i$ in $x$, we use the notation $x_{<i}$ to refer to $x_i$'s preceding sentences, that is, $x_{<i} = [x_1, x_2, \ldots, x_{i-1}]$. We define the context $c$ of $x_i$ to be a tuple of three elements. The first two are the entity $t$ of $x$ and the prefix $x_{<i}$. The third element is a relation triple $(s, r, o)$ extracted from $x_i$ using an information extraction (IE) system (Banko et al., 2007). We choose CompactIE (Bayat et al., 2022) because it is suitable for open domains and constructs triple elements from spans within the input sentence. Note that CompactIE is a small BERT model with 110M parameters that does not receive any other external information as input.

**Trigger**   To instantiate gLM.gen_sentence($c$), we format $c$ into a prompt. An example of our prompt is shown in Figure 1 (shortened for presentation purposes; the full prompt can be found in Appendix D). It includes $c$'s elements, such as the entity $t$ (William T. Freeman, a professor from MIT EECS), the prefix $\mathbf{x}_{<i}$, and the first two items of the relation triple ($s$: He and $r$: was born). These elements serve to constrain the scope of the output sentence $x_i'$. Importantly, we omit the third element $o$ of the relation triple, introducing a degree of freedom for gLM's generation process. The prompt then resembles a cloze test where the LM fills in the blank based on its internal knowledge (Petroni et al., 2019; Cohen et al., 2023a). To make gLM generate strictly one sentence, we change the system message and provide few-shot demonstrations. Given the prompt, gLM (in this case ChatGPT) successfully generates a sentence that aligns with the context, which is highlighted in red color. In Table 8 of Appendix B, we run alternative prompting strategies for gLM.gen_sentence, such as rephrasing (over-constraining) and asking for continuation (under-constraining). Our prompt significantly outperforms all baselines because it enforces an appropriate level of constraint.

**Detect**   We leverage a zero-shot prompt to instantiate aLM.detect($x_i, x_i', c$). This prompt includes the entity $t$, the prefix $\mathbf{x}_{<i}$, and the sentence pair ($x_i, x_i'$). We apply chain-of-thought prompting (Wei et al., 2022), asking aLM to provide first an explanation and then a conclusion. This enables the model to decompose the complex problem of contradiction detection into smaller reasoning steps. In practice, the generated explanation, together with $x_i$ and $x_i'$, can be returned to the user to improve the interpretability of the detection result. The binary conclusion (Yes or No) facilitates convenient parsing of the final detection result. An example of such a prompt is illustrated in Figure 2. The two gLM-generated sentences result in different birth years for William T. Freeman, clearly indicating contradiction. aLM (ChatGPT) correctly predicts the inconsistency for this case. According to Freeman's Wikipedia page (Wikipedia, 2023b), he was born in 1957, so both sentences are factually incorrect. In Table 9 of Appendix B, we examine other prompting strategies (Kojima et al., 2022; Wang et al., 2023) and find that chain-of-thought outperforms them.

**Revise**   To instantiate aLM.revise($x_i, x_i', c$), we stick to zero-shot prompting. Given the entity $t$ and the prefix $\mathbf{x}_{<i}$, our prompt instructs aLM to remove the conflicting information between $x_i$ and $x_i'$. We also provide specific instructions in the prompt such that the output is informative and coherent with the context. An example is shown in Figure 3. aLM (ChatGPT) successfully revises the sentence by eliminating the problematic birth date, and returns a faithful and fluent sentence.

**Generic Instantiation**   The only component specific to open-domain text generation within the prompts in Figures 1 to 3 is the first sentence (underlined in the figures). To adapt our approach to general tasks expressed by a free-form user-provided prompt $p$, we replace the sentence with a generic "Here is the start of an answer to the prompt $p$". In the case of question answering, $p$ takes the form of a question, such as "What is the birthplace of William T. Freeman?".

## 6   EXPERIMENTAL EVALUATION

In this section, we present an extensive evaluation of our work on open-domain text generation (Sections 6.1 and 6.2) and question answering (Section 6.3).

### 6.1   EXPERIMENTAL SETUP FOR OPEN-DOMAIN TEXT GENERATION

**Models**   We experiment with four popular instruction-tuned LMs: GPT-4 (gpt-4-0314) (OpenAI, 2023b), ChatGPT (gpt-3.5-turbo-0301) (OpenAI, 2023a), Llama2-70B-Chat (Touvron et al., 2023b), and Vicuna-13B (version 1.1) (Chiang et al., 2023). We evaluate these LMs as both gLM and aLM.

**Data and Annotation**   To construct evaluation data for open-domain text generation, we sample gLM to generate encyclopedic text descriptions for Wikipedia entities. Our primary test set, referred to as MainTestSet, consists of 360 descriptions spanning the four gLMs mentioned earlier and 30 diverse entities. For each gLM and each entity, we collect three different descriptions to enhance the reliability of our evaluation. In Table 2, we show that the generated descriptions are long, consisting of between 7.3 and 12.5 sentences on average. A sentence contains 1.6 fact triples on average. Table 2 also shows the average perplexity of the descriptions, measured using OPT-1.3B (Zhang et al., 2022).

The entities are carefully chosen from multiple Wikipedia-based sources (Lebret et al., 2016; Wikimedia, 2023; Wikipedia, 2023a) to capture different domains and varying levels of popularity. In Figure 4 of Appendix A, we list the 30 selected entities, highlight their diversity in domains and popularity, and detail our selection procedure.

Three authors (Ph.D. or Master's students in CS) perform human annotation to obtain the ground truth for `MainTestSet`. To improve knowledge coverage, our protocol instructs the annotators to review various online resources. Moreover, to ensure annotation quality, each sample undergoes independent annotation by two annotators. Then, they discuss to resolve mistakes and inconsistencies. We detail our annotation process in Appendix A. We achieve a Cohen's Kappa value of 88.9% and 82.7% for annotating self-contradictions, which indicates nearly perfect agreement (Cohen, 1960).

For validation, we use 12 entities. We compile a second test set called `2ndTestSet`, which contains a comprehensive set of 100 entities. For this set, we report the prediction results in Appendix B.

**Evaluation Steps and Metrics**   Our pipeline consists of triggering, detecting, and mitigating self-contradictions. To evaluate these steps, we leverage the following metrics:

- **Trigger**: We run Algorithm 1 to trigger self-contradictions. We calculate the frequency of self-contradictions: the ratio of sentences for which at least one self-contradiction is triggered. Then, we evaluate how self-contradiction complements retrieval-based methods (Shuster et al., 2021; Peng et al., 2023; Dinan et al., 2019; Min et al., 2023). To this end, we manually check if the contradictory information can be verified using Wikipedia or text obtained through web search. Our protocol for ensuring high-quality manual verification can be found in Appendix A.

- **Detection**: We run `aLM.detect` on the result obtained in the previous step. Then, we calculate and report standard classification precision (P), recall (R), and F1 score.

- **Mitigation**: We run Algorithm 3 on the `gLM`-generated text to obtain a revised version. To evaluate this step, we compare the original and the revised texts on the ratio of removed self-contradictions, informativeness, and fluency. When a sentence does not induce contradiction, we consider it as informative. The informativeness of a revised text is determined by comparing the number of informative sentences in the original text with the revised text and calculating the ratio. Note that this ratio might exceed 100% as our mitigation can produce new informative sentences. For fluency, we compute the increase in perplexity from the original text to the revised text.

**Sampling Temperature**   We leverage sampling for LM decoding, where temperature is an important parameter (Holtzman et al., 2020) and our evaluation involves multiple decision points for temperature. The ablation studies in Appendix B show that our results are robust to these choices.

We fix the temperature for the experiments in this section to investigate the effect of other variables. When generating the text descriptions, we use temperature 1.0 because it aligns with practical aspects of LM usage: (i) the default temperature in OpenAI API is 1.0 (OpenAI, 2023c), and (ii) users have the option to check multiple responses, highlighting the importance of sampling diversity (OpenAI, 2023a). When running `gLM.gen_sentence`, `aLM.detect`, and `aLM.revise`, we use temperature 0 for ChatGPT and GPT-4. This is because we desire maximum confidence for these functions. For Llama2-70B-Chat and Vicuna-13B, we have to use temperature 1.0 to avoid repetitive text.

## 6.2   EVALUATION RESULTS FOR OPEN-DOMAIN TEXT GENERATION

**Self-contradictions are Significant**   Table 2 highlights the significance of self-contradictions. First, our approach triggers a prevalence of self-contradictions, ranging from 15.7% to 22.9% of all sentences depending on the gLM. The more powerful the model, the fewer self-contradictions it generates. For instance, GPT-4 has the lowest ratio of self-contradictions. In Figure 5 of Appendix B, we show that these self-contradictions spread across popular and lesser-known entities, instead of concentrating on a small subset of entities. Moreover, a substantial portion of these self-contradictions (e.g., 35.2% for ChatGPT) cannot be verified using online text, making them highly unlikely to be resolved by retrieval-based techniques (Shuster et al., 2021; Peng et al., 2023). Thus, our work stands as a valuable complement to retrieval-based methods. Vicuna-13B's self-contradictions are often about basic facts that are easily verifiable online, while other LMs produce more nuanced information.

Table 2: Statistics of generated open-domain text descriptions and triggered self-contradictions, for different gLMs on `MainTestSet`.

| gLM | avg. num. sentences | perplexity | self-contra. triggered (↓) | unverifiable online |
|---|---|---|---|---|
| GPT-4 | 12.5 | 7.02 | 15.7% | 27.5% |
| ChatGPT | 9.6 | 5.72 | 17.7% | 35.2% |
| Llama2-70B-Chat | 11.3 | 5.44 | 19.0% | 30.1% |
| Vicuna-13B | 7.3 | 6.82 | 22.9% | 20.5% |

Table 3: Results of using the same aLM (ChatGPT in this case) to detect and mitigate self-contradictions produced by different gLMs on `MainTestSet`.

| gLM | Detection | | | Mitigation | | |
|---|---|---|---|---|---|---|
| | P (↑) | R (↑) | F1 (↑) | self-contra. removed (↑) | informative facts retained (↑) | perplexity increased (↓) |
| GPT-4 | 80.1% | 79.7% | 79.9% | 76.3% | 99.9% | 0.67 |
| ChatGPT | 84.2% | 83.2% | 83.7% | 89.5% | 100.8% | 0.44 |
| Llama2-70B-Chat | 72.9% | 89.0% | 80.2% | 85.4% | 95.5% | 0.86 |
| Vicuna-13B | 74.4% | 83.2% | 78.6% | 82.7% | 95.1% | 1.78 |

Table 4: Results of using different aLMs to detect and mitigate self-contradictions produced by the same gLM (ChatGPT in this case) on `MainTestSet`.

| aLM | Detection | | | Mitigation | | |
|---|---|---|---|---|---|---|
| | P (↑) | R (↑) | F1 (↑) | self-contra. removed (↑) | informative facts retained (↑) | perplexity increased (↓) |
| GPT-4 | 91.3% | 82.1% | 86.5% | 79.7% | 97.7% | 0.52 |
| ChatGPT | 84.2% | 83.2% | 83.7% | 89.5% | 100.8% | 0.44 |
| Llama2-70B-Chat | 95.3% | 45.8% | 61.9% | 44.4% | 98.8% | 0.22 |
| Vicuna-13B | 90.0% | 25.1% | 39.3% | 25.5% | 100.2% | 0.28 |

**Our Detection and Mitigation Methods are Effective** Next, we instantiate our method with ChatGPT as aLM. The results in Table 3 show that our approach is consistently effective across different gLMs. The detection achieves a high F1 score of ∼80%. The mitigation is able to remove a significant portion of self-contradictions (76.3% to 89.5%) without harming text fluency (perplexity increase is small). It also maintains informativeness, i.e., the number of informative facts is maintained.

**Performance Varies across LMs** In Table 4, we assess different aLMs in detecting and mitigating self-contradictions produced by the same gLM, ChatGPT. The results reveal that both proprietary LMs (GPT-4 and ChatGPT) have strong performance. Open-source LMs (Llama2-70B-Chat and Vicuna-13B) show limitations, particularly in low detection recall and inability to effectively remove self-contradictions. Fortunately, state-of-the-art open-source instruction-tuned LMs have substantially advanced in recent months, with Llama2-70B-Chat (released in late July 2023, current state-of-the-art) significantly outperforming Vicuna-13B (version 1.1, released in early May 2023 and then state-of-the-art according to LMSYS (2023b)). We anticipate that this positive trend will continue.

**Efficiency Analysis** For a text $\mathbf{x} = [x_1, x_2, \ldots, x_{|\mathbf{x}|}]$, our method makes a constant number of LM queries for each sentence $x_i$ and the total number of LM queries grows linearly with $|\mathbf{x}|$. For each $x_i$, we include its prefix $\mathbf{x}_{<i}$ in our prompts. Thus, the total length of the prompts grows quadratically with $|\mathbf{x}|$. In our experiments, we monitor the number of tokens consumed by the OpenAI API, which reflects both speed and compute in practice. For ChatGPT, generating a text description for an entity costs 259 tokens on average. Detecting and mitigating self-contradictions come with a cost of 79x and 90x, respectively. Considering the price of 0.002$ per 1k tokens, the expense of detection and mitigation amounts to 0.04$ and 0.05$, respectively. We consider these costs acceptable, especially for scenarios where a high level of trustworthiness is demanded.

Table 5: Question answering using the PopQA benchmark (Mallen et al., 2023) with aLM ChatGPT.

| gLM | Vanilla | | Retrieval Augmented | |
|---|---|---|---|---|
| | self-contra. predicted (↓) | P (↑) | self-contra. predicted (↓) | P (↑) |
| ChatGPT | 18.2% | 83.2% | 12.7% | 83.8% |
| Llama2-70B-Chat | 38.0% | 79.6% | 21.5% | 74.2% |

**More Details in Appendix**  In Appendix B, we provide additional experimental results for open-domain text generation. We first show that self-contradiction is common across different entities. Next, we show that our detection method outperforms SelfCheckGPT (Manakul et al., 2023b). Moreover, we perform ablation studies to show the robustness of our results across different temperatures and to validate our design decisions discussed in Sections 4 and 5. Finally, we explore the relationship between the choice of gLM and aLM, and showcase the usefulness of our pipeline on 2ndTestSet.

In Appendix C, we provide more real-world examples on the sentence level and the result of the method on full-text examples. Further, we showcase a self-contradiction that cannot be verified using online text, and a failure case of aLM.detect where aLM is Vicuna-13B.

## 6.3 EXPERIMENTS ON QUESTION ANSWERING

We adapt our approach to question answering as described at the end of Section 5. We focus on 1.5K randomly sampled questions from PopQA (Mallen et al., 2023). We perform experiments in two settings, vanilla generation, and retrieval augmentation, where generation and detection are provided with adaptively retrieved text snippets as proposed by Mallen et al. (2023). We use ChatGPT as aLM and the default temperatures (see the end of Section 6.1). To compute the precision of our detection method, we manually annotate all sentence pairs that are predicted to be contradictory.

The results for the two evaluated gLM, ChatGPT and Llama2-70B-Chat, are presented in Table 5. Even though self-contradictions decrease in the retrieval augmented setting by around 30% in ChatGPT and almost 50% in Llama2-70B-Chat, a significant amount of self-contradictions (12.7% to 38.0%) is detected in all settings with high precision (74.2% to 83.8%). This implies that even with retrieval augmentation, at least 10.6% of ChatGPT and 16.0% of Llama2-70B-Chat generated answers contain self-contradictions. A qualitative analysis shows that self-contradictions often arise when the retrieved knowledge is irrelevant to the posed question or lacks the crucial information for a correct answer. This reaffirms our claim that our method serves as a valuable complement to retrieval-based methods.

## 7 CONCLUSION AND DISCUSSION

In this work, we presented a comprehensive investigation into self-contradiction, an important form of hallucination produced by LMs. Self-contradictions arise when the LM generates two contradictory sentences within the same context, thereby revealing the LM's lack of factuality. To address this issue, we developed an algorithm to trigger, detect, and mitigate self-contradictions. Importantly, our approach is built upon prompting strategies, making it applicable to black-box LMs without requiring retrieval of external knowledge. We conducted an extensive evaluation on four modern instruction-tuned LMs and two different tasks, demonstrating the benefits of our approach: it effectively exposes, accurately detects, and appropriately mitigates self-contradictions.

Our framework currently handles the contradictions of two sentences sampled at the same text position. A future work item is to efficiently detect contradictions across different positions. Moreover, our approach serves as a post-processing method on LMs' output text. It is open to explore the potential of fine-tuning smaller open-source LMs to achieve high performance as aLM or even training LMs to avoid generating contradictory content. Finally, we note that other kinds of hallucinations than self-contradictions exist. For example, LMs can consistently generate incorrect content. As self-contradictions are prevalent and our approach complements retrieval-based augmentation methods at hallucination detection, we believe that this work is significant and important for understanding and enhancing the trustworthiness of modern LMs.

## REPRODUCIBILITY STATEMENT

All our work is open source and available at `https://github.com/eth-sri/chatprotect`. The experiments were conducted against LMs that are publicly available, either fully open-source through model parameters or via public APIs and we set the random seeds where possible to ensure reproducibility. Moreover, both in the main paper and in the Appendix, we give details on the experimental setup used, and how and which hyperparameters were selected.

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

## A    MORE DETAILS ON EXPERIMENTAL SETUP

**Selecting Diverse and Representative Entities**    To construct our test sets, we select entities from Wikipedia titles. We carefully assemble the test sets to ensure the diversity and representativeness of the entities. The majority of entities are randomly sampled from the list of human biographies in WikiBio (Lebret et al., 2016) and the list of all Wikipedia articles (Wikimedia, 2023) (mostly non-human). This strikes a balance between human and non-human entities. We also randomly sample from the list of featured articles (Wikipedia, 2023a) to include more popular entities. After sampling, we make sure that the selected entities cover different degrees of popularity and different domains. To measure the level of popularity, we quote the entire phrase of each entity and search it on Google, which returns webpages that match the exact phrase. Moreover, we do not include entities where the evaluated LMs refuse to provide any useful information or return irrelevant text.

The 30 entities in `MainTestSet` and their statistics are presented in Figure 4. The 100 entities in `2ndTestSet` and their popularity are listed in Figures 7 and 8. Our validation set consists of 12 entities: `Angela Merkel`, `Augustin-Louis Cauchy`, `Eike von Repgow`, `Hernán Siles Zuazo`, `Marbel Arch`, `Marillac College`, `Mikhail Bulgakov`, `Okolo N'Ojie`, `Ruy de Sequeira`, `Shahar Pe'er`, `Spain`, `Tan Chorh Chuan`.

**Details on Human Annotation Process**    Our human annotation process aims to derive three labels: self-contradiction, factuality, and verifiability of contradictory information. We are aware that annotating these labels is a challenging task, even for human beings. To enhance annotation quality, we ensure that each annotation decision is a joint effort of two annotators. Specifically, we have three annotators in total. One annotator annotates the entire dataset and the other two annotate two non-overlapping splits. After that, the annotators discuss together to fix mistakes and resolve inconsistencies. We find that this discussion step greatly boosts annotation quality. After the discussion, a small number of disagreements remain due to their subjective nature and we use the first annotator's label as the final decision.

While the annotation of self-contradictions does not require external knowledge, annotating the verifiability of contradictory information requires the annotators to thoroughly scan the Internet to obtain all useful information. Since our test entities are all from Wikipedia, we first instruct the annotators to diverse Wikipedia articles: (i) articles of all entities in the contradictory sentences; (ii) articles of relevant entities mentioned in (i); (iii) If non-English articles provided more information than English counterparts, the annotators used Google Translate to review non-English versions translated to English. We find that Wikipedia has a high coverage of necessary information for verification, which is also supported by a study by Min et al. (2023) (Appendix A.5 in their paper). For the minority of cases where Wikipedia is insufficient, we ask the annotators to search the Internet extensively for important keywords such as mentions of entities and their relationships. Note that the purpose of this annotation is to evaluate how our work complements retrieval-based methods. Therefore, the annotators focus only on content that can be handled by existing retrieval-based methods (Shuster et al., 2021; Peng et al., 2023; Dinan et al., 2019; Min et al., 2023), excluding considerations of figures, images, files, etc.

To annotate factuality, we ask the annotators to check all relevant online resources they could find. If a piece of information can be supported by text on the Internet, we mark a sentence as factual. Otherwise, we label it as non-factual.

**Compute**    For ChatGPT and GPT-4, we query the OpenAI API (OpenAI, 2023c). We run Vicuna-13B with the FastChat API (LMSYS, 2023a) on NVIDIA A100 80GB GPUs. We use the service of Together AI (TogetherAI, 2023) for running Llama2-70B-Chat.

## B    MORE EXPERIMENTS ON OPEN-DOMAIN TEXT GENERATION

**Distribution of Self-contradictions**    To provide insight into the distribution of self-contradictions, Figure 5 provides a breakdown of self-contradictions generated by ChatGPT among individual entities. It shows that self-contradiction is a common issue among different entities and popularity.

**Comparison with SelfCheckGPT**   We compare our method with SelfCheckGPT (Manakul et al., 2023b) on the task of detecting self-contradictions. SelfCheckGPT (Manakul et al., 2023b) leverages three black-box methods for detecting hallucinations. They are based on unigram probabilities, BertScore (Zhang et al., 2020), and multiple-choice question answering and generation (MQAG) (Manakul et al., 2023a). We perform a thorough comparison between our prompting-based detection approach with these three methods in two settings: (a) our setting of detecting self-contradictions of sentence pairs, and (b) their setting of detecting non-factual sentences given 20 alternative samples. We use ChatGPT as both gLM and aLM. To enable setting (b), we annotate the factuality of each generated sentence, sample 20 alternative sentences, and adapt our prompt in Figure 2. We change our prompts to output a score ranging from 0 to 10. Our prompt adapted for setting (b) can be found in Figure 16. The results are plotted in Figure 6 and show that our approach substantially outperforms all three variants of SelfCheckGPT, especially in our setting of detecting self-contradictions.

**Robustness across Different Temperature Values**   We have tested temperature values 0, 0.25, 0.5, 0.75, and 1.0 for both gLM and aLM (ChatGPT for both cases in this experiment). The results show that the claims of our paper hold across different temperature values: (i) self-contradictions are consistently prevalent; (ii) our approach for addressing self-contradictions is robust.

We first fix the temperature of aLM to the default value 0 and vary the temperature of gLM in generating the initial text. For each temperature, gLM produces new text descriptions, which require new expensive human annotations. To make human annotation feasible, we annotate all sentence pairs predicted by our detection method to be contradictory and report detection precision. The results are presented in Table 6. We find that both the ratio of sentences predicted to be self-contradictory and the precision of our detection are consistent across different temperatures.

Then, we fix the temperature of gLM to the default values and vary the temperature of aLM in detecting self-contradictions. We reuse existing human annotations in this experiment. The results are presented in Table 7, showing that the precision, recall, and F1 scores have very small variance.

**Ablation Study on Trigger**   We consider three alternative prompting strategies as baselines for gLM.gen_sentence: (i) *Continue*: we directly ask gLM to generate a continuation of $\mathbf{x}_{<i}$ as $x_i'$. This method imposes insufficient constraints on $x_i'$, often resulting in non-contradictions where $x_i$ and $x_i'$ refer to different subjects. (ii) *Rephrase*: we provide gLM with $\mathbf{x}_{<i}$ and $x_i$, and request gLM to rephrase $x_i$ to $x_i'$. This strategy overly restricts $x_i'$ because rephrasing inherently preserves the semantic meaning of $x_i$. (iii) *Q&A*: we provide gLM with $\mathbf{x}_{<i}$ and $x_i$, prompting gLM to generate questions regarding $x_i$. We then instruct gLM to answer the questions. With this prompt, the level of constraint is decided by gLM and is not controlled by us. We compare these alternatives with our prompt in Table 8. The results clearly show that our prompt outperforms the baselines, by triggering more self-contradictions and leading to higher detection precision. This success can be attributed to the appropriate level of constraint our prompt enforces on the generation of $x_i'$.

**Ablation Study on Detection**   For aLM.detect, we compare our prompting strategy in Figure 2 with various baselines in Table 9. The first baseline involves a straightforward prompt which *directly asks* aLM to answer Yes/No on whether a sentence pair is contradictory based on the context. This baseline is inaccurate and cannot provide interpretability of the decision-making process. Following Kojima et al. (2022), we prompt aLM to reason in a *step-by-step* manner with a single query that elicits the model to print reasoning steps before coming to a definite conclusion. We find that step-by-step reasoning is not well applicable to contradiction prediction, where asking for concrete steps often leads the model to follow wrong reasoning paths. On the contrary, our chain-of-thought prompting strategy (Wei et al., 2022) asks the model in two turns to first provide a free-form explanation only and then conclude with Yes/No, which boosts accuracy. We also explore the use of *multi-path* reasoning (Wang et al., 2023), which extends over chain-of-thought prompting by querying aLM to provide multiple explanations before concluding via majority-vote. We find that multi-path reasoning does not provide benefits over our prompt, while drastically increasing the cost.

**Ablation Study on Mitigation**   In Table 10, we present a breakdown of the results obtained from running our iterative mitigation procedure in Algorithm 3. Each iteration progressively removes self-contradictions. After the third iteration, we remove the sentences corresponding to the remaining small amount of predicted self-contradictions. This removal preserves fluency and informativeness.

In contrast, a baseline method that *naively drops* all predicted self-contradictions from the beginning results in significantly lower informativeness and fluency.

**Relationship between** `gLM` **and** `aLM`   In Section 6.2, we present results on the performance of each LM as either `gLM` or `aLM`. We now further explore the performance between different combinations of `gLM` and `aLM`. To this end, we try all possible combinations of GPT-4 and ChatGPT as `gLM` and `aLM`. The results are presented in Table 11. We draw two observations. First, combining different `gLM` and `aLM` achieves a higher F1 score than using the same LM, underscoring the benefit of cross-examination (Cohen et al., 2023b). Second, content generated by GPT-4 is more difficult to classify: in terms of F1 score, ChatGPT-ChatGPT outperforms GPT-4-ChatGPT and ChatGPT-GPT-4 outperforms GPT-4-GPT-4. We find that the differences in the sentence pairs generated by GPT-4 are often more nuanced and harder to identify.

**Results on** `2ndTestSet`   We now discuss the results of running our approach on `2ndTestSet`. Given the large size of `2ndTestSet`, we do not perform human annotation and only report prediction results. In this series of experiments, both `gLM` and `aLM` are ChatGPT. Overall, 12.7% of the sentences are predicted to be self-contradictory. The breakdown is plotted in Figures 7 and 8, showing again that self-contradiction is a common issue across entities. The ratio of detected self-contradictions negatively correlates with entity popularity. Our mitigation successfully removes all predicted self-contradictions, while maintaining informativeness (we retain 100.9% sentence pairs that are predicted to be non-contradictory) and fluency (perplexity increase is only 0.42).

## C   MORE REAL-WORLD EXAMPLES

Next, we provide more long examples for a better understanding of the effectiveness of our approach.

**Sentence-level Examples**   In Table 12, we present additional examples of self-contradictory sentence pairs. We can see that these LMs produce diverse hallucinated information, such as geographical location and wedding date. For all cases, our mitigation successfully eliminates self-contradiction and generates a coherent revision. It may happen that the first sentence in a pair is factually correct, but the second one is factually wrong, e.g., the example for the entity "`Black Mirror`". Such cases reflect LMs' internal uncertainty, which is removed by our method to improve certainty and factuality.

**Text-level Examples**   In Figures 9 to 11, we show text-level examples. Each example consists of the original text and the revised text for a given entity. The examples are generated by different `gLMs` and revised by the same `aLM` (i.e., ChatGPT). Our method applies mitigation locally on sentences where self-contradictions are detected, keeping other sentences unchanged. Enforcing local changes is important for maintaining the overall fluency and informativeness of the text. Our mitigation successfully removes self-contradictory information in all but one case (the purple sentence in Figure 10). In that case, the revised sentence has the same semantic meaning as the original sentence.

**A Self-contradiction Unverifiable using Online Text**   In Table 13, we show an example of self-contradiction for which neither the corresponding Wikipedia page nor results from web search contain relevant information. When prompting ChatGPT as `aLM`, our method detects this self-contradiction and decides to remove the sentence, which improves the overall factuality.

**Failure Case of Vicuna-13B**   The detection recall of Vicuna-13B on `aLM.detect` is far lower than other considered LMs. To understand the reason, we inspect false negatives of Vicuna-13B and compare its decision with ChatGPT's. We find that Vicuna-13B usually struggles to identify conflicting information. As a result, it tends to make up erroneous explanations that lead to false negatives. We provide such a case in Figure 12, where ChatGPT outputs the correct response in a succinct and confident manner. In contrast, Vicuna-13B makes up three flawed reasons.

## D   COMPLETE PROMPTS

We now provide the full prompts of our method and the ablation baselines.

**Prompts of Our Method**    The full prompts of our method are shown in Figures 13 to 15. They correspond to the shortened versions in Figures 1 to 3. To ensure that gLM generates strictly one sentence in Figure 13, we change the system message and provide the LM with three-shot demonstrations using entities "`Douglas Adams`", "`Kayne West`", and "`Angela Merkel`".

**Our Prompt Adapted for Detecting Non-factuality**    To compare with SelfCheckGPT (Manakul et al., 2023b) in Figure 6b, we adapt our prompt for `aLM.detect` to the task of predicting if a sentence is non-factual given 20 alternative samples. The adapted prompt is shown in Figure 16. It first asks the model if there are contradictions between the original sentence and the alternatives. Then, it queries the model to conclude whether the original sentence is factually correct.

**Prompts of Ablation Baselines**    In Figures 17 to 19, we provide the prompts for the baselines for `gLM.gen_sentence` in Table 8. For "Continue" (Figure 17) and "Q&A" (Figure 19), we generate two alternative sentences to match the number of sentence pairs produced by our method. For "Continue" (Figure 17) and "Rephrase" (Figure 18), we use the same three-shot demonstrations as our method in Figure 13. The prompts of the baselines for `aLM.detect` in Table 9 are presented in Figures 20 to 22. The baseline "Multi-path" in Figure 22 uses the same prompt as our method in Figure 14. The difference is that "Multi-path" samples 5 explanations with temperature 1, while our method generates one explanation with temperature 0.

Homer Simpson, George Whitefield, John Francis Daley, Perseus (constellation), Mick Garris, Love, Inc. (TV Series), Lütfi Elvan, John Holt (author), Diane Arkenstone, Joe Ledley, Ke Apon Ke Por, Mike Rawlings, Fernand Seguin, Ronald Ridenhour, English Arts and Crafts architecture, Corey Parker (actor), Auguste Clésinger, Wen Tianxiang, Gordon Gollob, 2003 World Judo Championships - Men's 60 kg, Hellwegbörden, Shlomo Molla, Thirty-First Army (Japan), Prince Albert of Saxe-Altenburg, Brazil-Angola relations, Bailey-Thompson House, Ecologist Party for the Development of Burkina, Pattimura Stadium, C. T. Blackfan, 1990-91 Austrian Hockey League season

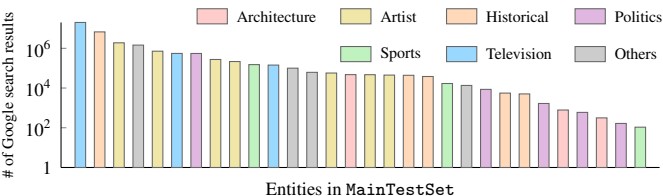

Figure 4: The entities in MainTestSet (left). They have different levels of popularity according to the number of Google research results with exact phrase match (the bar chart). They also cover different domains (indicated by the different colors).

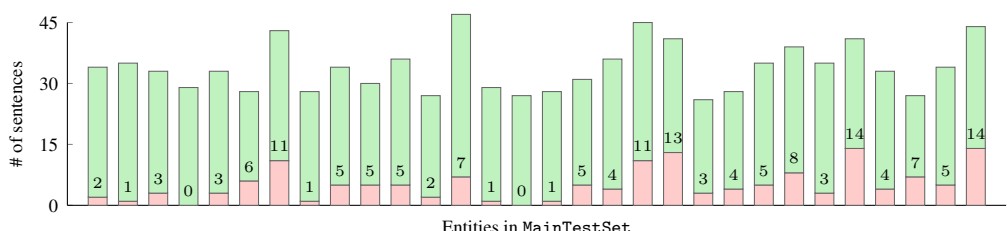

Figure 5: Breakdown of ChatGPT-generated self-contradictions (red) and non-contradictory sentences (green). The entities are sorted by their popularity, i.e., the same order as in Figure 4.

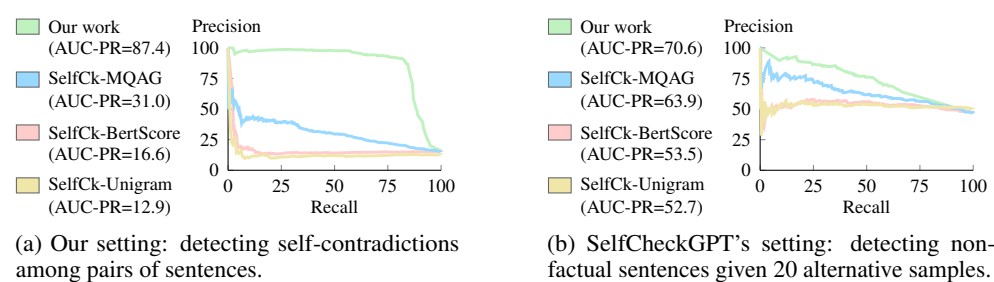

(a) Our setting: detecting self-contradictions among pairs of sentences.

(b) SelfCheckGPT's setting: detecting non-factual sentences given 20 alternative samples.

Figure 6: Comparison between our work and SelfCheckGPT (Manakul et al., 2023b) in two settings.

Table 6: Different choices of temperature values for `gLM` during text generation.

| temperature | self-contra. predicted | P (↑) |
|---|---|---|
| 0 | 17.4% | 84.1% |
| 0.25 | 17.8% | 81.3% |
| 0.5 | 17.2% | 79.2% |
| 0.75 | 16.6% | 83.3% |
| 1 | 18.3% | 83.4% |

Table 7: Different choices of temperature values for `aLM.detect`.

| temperature | P (↑) | R (↑) | F1 (↑) |
|---|---|---|---|
| 0 | 84.3% | 83.8% | 84.0% |
| 0.25 | 83.4% | 84.4% | 83.9% |
| 0.5 | 83.3% | 83.8% | 83.6% |
| 0.75 | 80.9% | 82.7% | 81.8% |
| 1 | 84.2% | 83.2% | 83.7% |

Table 8: Ablation study on `gLM.gen_sentence`.

| | self-contra. triggered (↑) | P (↑) |
|---|---|---|
| Continue | 5.1% | 76.8% |
| Rephrase | 0.2% | 33.3% |
| Q&A | 10.3% | 81.9% |
| Ours | 17.7% | 84.2% |

Table 9: Ablation study on `aLM.detect`.

|  | P (↑) | R (↑) | F1 (↑) |
|---|---|---|---|
| Directly ask | 83.0% | 65.4% | 73.1% |
| Step-by-step (Kojima et al., 2022) | 89.0% | 54.2% | 67.4% |
| Multi-path (Wang et al., 2023) | 86.9% | 77.7% | 82.0% |
| Ours | 84.2% | 83.2% | 83.7% |

Table 10: Ablation study on our mitigation algorithm in Algorithm 3.

|  | self-contra. removed (↑) | informative facts retained (↑) | perplexity increased (↓) |
|---|---|---|---|
| Original | 0% | 100.0% | 0 |
| Iteration 1 | 69.3% | 101.6% | 0.42 |
| Iteration 2 | 74.5% | 102.2% | 0.45 |
| Iteration 3 | 75.8% | 102.3% | 0.46 |
| Our final result | 89.5% | 100.8% | 0.44 |
| Naively drop | 94.8% | 89.4% | 0.72 |

Table 11: The relationship between `gLM` and `aLM` for `aLM.detect`.

| gLM | aLM | P (↑) | R (↑) | F1 (↑) |
|---|---|---|---|---|
| ChatGPT | ChatGPT | 84.2% | 83.2% | 83.7% |
| ChatGPT | GPT-4 | 91.3% | 82.1% | 86.5% |
| GPT-4 | ChatGPT | 80.1% | 79.7% | 79.9% |
| GPT-4 | GPT-4 | 88.3% | 65.7% | 75.3% |

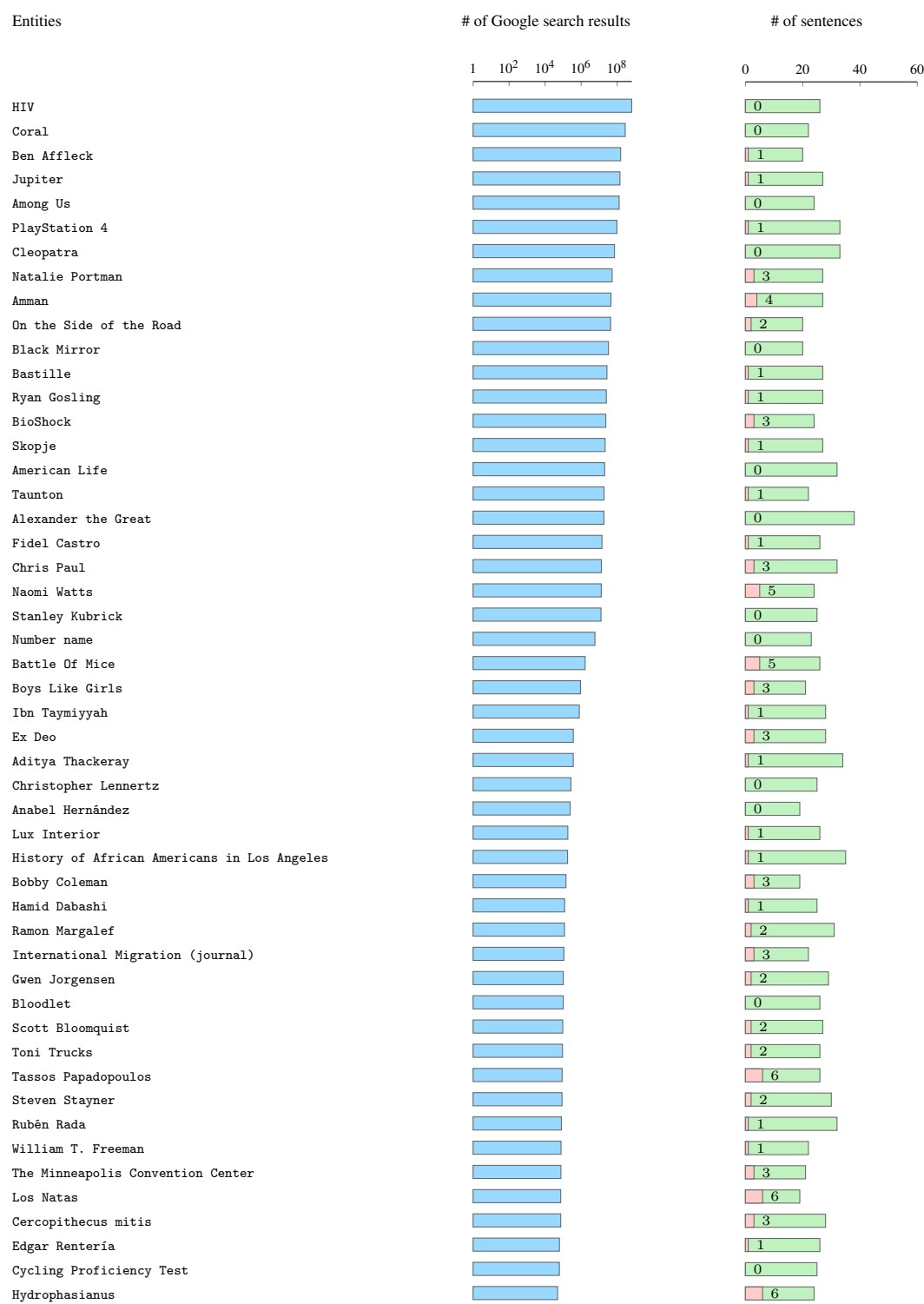

Figure 7: The first 50 entities in our 2ndTestSet. The second column shows their popularity. The third column shows the number of sentences predicted to be self-contradictory or non-contradictory, when we use ChatGPT as both gLM and aLM.

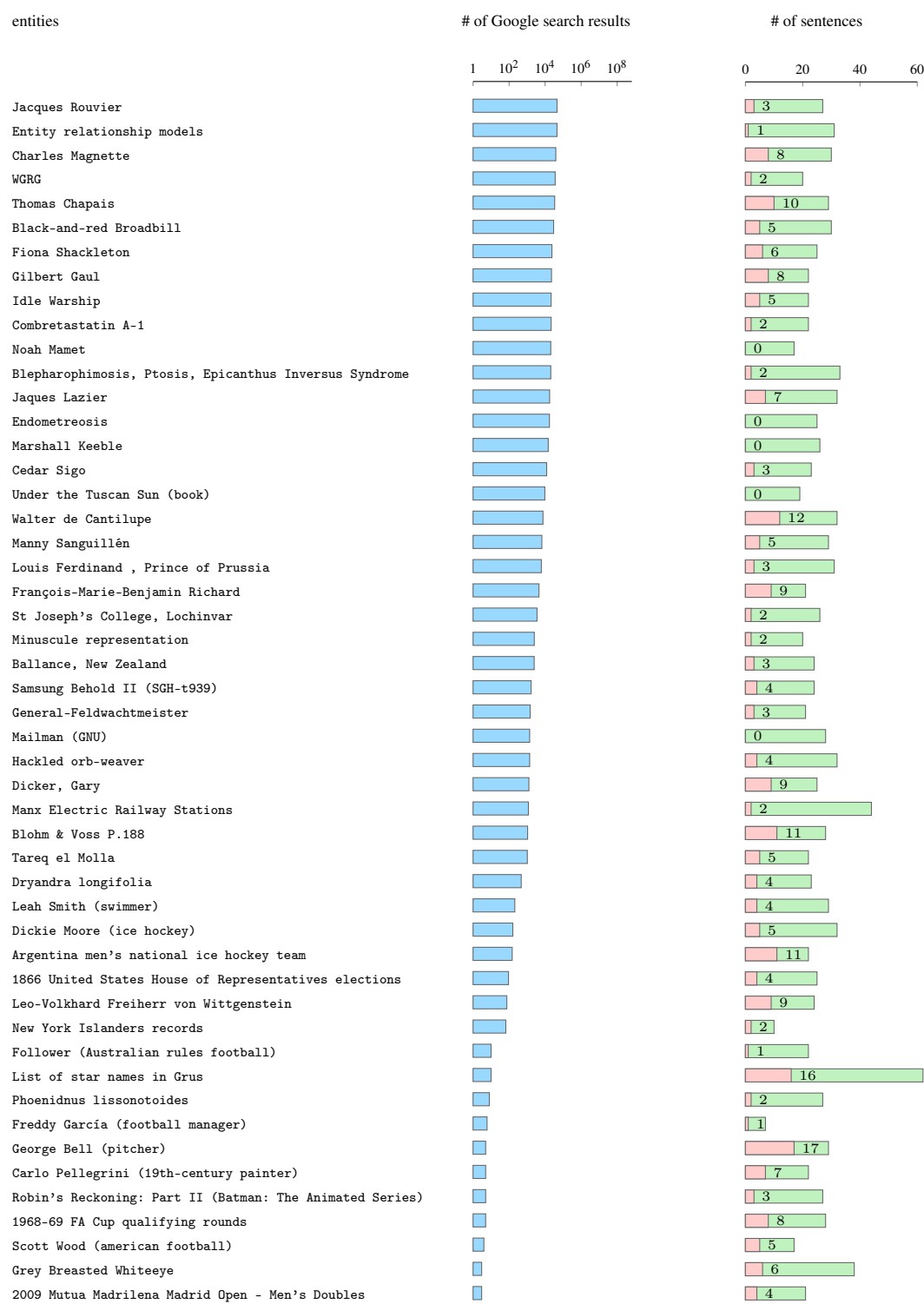

Figure 8: The second 50 entities in our 2ndTestSet. The second column shows their popularity. The third column shows the number of sentences predicted to be self-contradictory or non-contradictory, when we use ChatGPT as both gLM and aLM.

Table 12: Real-world self-contradictory hallucinations generated by different gLMs (left-most column). Sentences marked with green color (resp., red color) are factually correct (resp., wrong). We leverage ChatGPT as aLM for detection and mitigation.

| | | |
|---|---|---|
| GPT-4 | [Prefix] | Natalie Portman, born on June 9, 1981, in Jerusalem, Israel, is an actress, producer, and director. She is best known for her roles in films such as "Black Swan," "Star Wars," and "V for Vendetta." With dual American and Israeli citizenship, Portman began her acting career at a young age after being discovered by a talent agent in a pizza parlor. Portman's film debut was in Luc Besson's 1994 film "The Professional," where she played a young girl mentored by a hitman. Later, she gained international fame for her role as Padma Amidala in the "Star Wars" prequel trilogy, starting with "Star Wars: Episode I - The Phantom Menace" in 1999. Her performance in "Black Swan" (2010) won her several awards, including an Academy Award for Best Actress. In addition to her acting career, Portman is a dedicated supporter of numerous social and political causes, notably advocating for animal rights, environmental issues, and women's rights. She is also a Harvard-educated psychologist, having earned her bachelor's degree in psychology in 2003. |
| | [Sentence pair] | In 2011, Portman married French dancer and choreographer Benjamin Millepied. Portman married Benjamin Millepied in 2012. |
| | [Mitigation] | Portman married Benjamin Millepied. |
| GPT-4 | [Prefix] | Francois-Marie-Benjamin Richard (1819-1908) was a French Roman Catholic clergyman who served as the Archbishop of Paris from 1886 to 1908. He was born on March 1, 1819, in Nantes, France, and was ordained a priest in 1845. During his early career, Richard served as a parish priest and a vicar general in the Diocese of Nantes. He was appointed the Bishop of Belley in 1872 and held that position until 1886. On November 3, 1886, he was appointed the Coadjutor Archbishop of Paris, and a few months later, on January 6, 1887, he became the Archbishop of Paris following the death of Archbishop Joseph-Hippolyte Guibert. |
| | [Sentence pair] | He was made a cardinal by Pope Leo XIII in 1890. He was made a cardinal in 1896. |
| | [Mitigation] | He was made a cardinal by Pope Leo XIII. |
| ChatGPT | [Prefix] | Skopje is the capital and largest city of North Macedonia. |
| | [Sentence pair] | It is located in the central part of the country, on the Vardar River. It is located in the northern part of the country, on the Vardar River. |
| | [Mitigation] | It is located on the Vardar River. |
| ChatGPT | [Prefix] | Louis Ferdinand von Preussen (1907-1994) was a member of the House of Hohenzollern and the head of the Prussian royal House from 1951 until his death. He was the eldest son of Wilhelm, Crown Prince of Germany, and was born in Potsdam, Germany. During World War II, Louis Ferdinand served in the German army as a tank commander and was wounded twice. After the war, he became a strong advocate for a united Germany. He was also an active supporter of various charitable causes and founded the Prince Louis Ferdinand Foundation to promote cultural and historical preservation in Germany. |
| | [Sentence pair] | Louis Ferdinand was married to Grand Duchess Kira of Russia, and they had four children. Louis Ferdinand, Prince of Prussia, had six children. |
| | [Mitigation] | Louis Ferdinand was married to Grand Duchess Kira of Russia. |
| Llama2-70B -Chat | [Prefix] | Pattimura Stadium is a multi-purpose stadium located in Makassar, Indonesia. It is named after Pattimura, a National Hero of Indonesia who fought against the Dutch colonial rule in the 19th century. The stadium was built in 1967 and has a seating capacity of approximately 20,000 spectators. Pattimura Stadium is primarily used for football matches, and it has hosted numerous domestic and international football tournaments. |
| | [Sentence pair] | The stadium is home to the Indonesia national football team and has also hosted concerts, festivals, and other events. Pattimura Stadium is home to the Makassar football club and has also hosted several international concerts and events. |
| | [Mitigation] | The stadium has also hosted concerts, festivals, and other events. |
| Llama2-70B -Chat | [Prefix] | Ron Ridenhour was a former US Army soldier who became a prominent anti-war activist and whistleblower during the Vietnam War. He is best known for his role in exposing the My Lai Massacre, a mass killing of hundreds of unarmed Vietnamese civilians by US soldiers in 1968. Ridenhour was born in 1946 in Louisiana and enlisted in the Army in 1966. |
| | [Sentence pair] | He was deployed to Vietnam in 1968 and served as a door gunner on a helicopter. He was deployed to Vietnam in 1967. |
| | [Mitigation] | He served as a door gunner on a helicopter. |
| Vicuna-13B | [Prefix] | Black Mirror is a British science fiction anthology television series created by Charlie Brooker. |
| | [Sentence pair] | The series debuted on Channel 4 in 2011 and later moved to Netflix for subsequent seasons. The series debuted on Netflix in 2011. |
| | [Mitigation] | The series moved to Netflix for subsequent seasons. |
| Vicuna-13B | [Prefix] | American Life is a Yes song from their album of the same name, released in 1986. The song is about the stress of modern life in the United States and the longing for a simpler, more fulfilling existence. |
| | [Sentence pair] | The song reached number 9 on the Billboard Hot 100 chart. The song reached number 21 on the Billboard Hot 100 chart. |
| | [Mitigation] | The song charted on the Billboard Hot 100. |

Thomas Chapais was a Canadian lawyer, historian, and politician born on December 24, 1858 in Saint-Denis-de-la-Bouteillerie, Quebec. He was a member of Canadian parliament for Champlain constituency from 1896 to 1904. Chapais was also a prolific writer and historian. He wrote several books on Canadian history, including "The Great Intendant: A Chronicle of Jean Talon in Canada, 1665-1672" and "La Bourgade de Quebec et son faubourg Saint-Jean". He was also a member of the Royal Society of Canada and served as its president from 1910 to 1911. Chapais was a prominent figure in the formation of the Quebec nationalist movement. He argued that French Canadians needed to assert their cultural and political rights within Canada. In 1899, he was a co-founder of the Ligue nationaliste canadienne, an organization that sought to promote French Canadian nationalism. Thomas Chapais died on August 17, 1946 in Quebec City at the age of 87.

Thomas Chapais was born in Saint-Denis-de-la-Bouteillerie, Quebec. He was a member of Canadian parliament for Champlain constituency from 1896 to 1904. Chapais was also a prolific writer and historian. He wrote several books on Canadian history, including "The Great Intendant: A Chronicle of Jean Talon in Canada, 1665-1672" and "La Bourgade de Quebec et son faubourg Saint-Jean". He was a member of the Royal Society of Canada. Chapais played a role in the Quebec nationalist movement. He argued that French Canadians needed to assert their cultural and political rights within Canada. In 1899, he was a co-founder of the Ligue nationaliste canadienne, an organization that sought to promote French Canadian nationalism.

Figure 9: The original text and the revised text for the entity "Thomas Chapais". We leverage ChatGPT as both gLM and aLM. Red color represents sentences removed from the original text, while other colors represent sentences revised to mitigate self-contradictions.

Los Natas is an Argentine stoner rock band that was formed in 1993 in Buenos Aires. The band consists of Sergio Chotsourian (guitar and vocals), Claudio Filadoro (bass), and Walter Broide (drums). Their music combines elements of stoner rock, doom metal, psychedelic rock, and heavy rock with hints of Argentine folk music. The band's name, "Los Natas," is derived from the word "satan" spelled backward ("natas" in Spanish) which showcases the band's interest in the dark and mystical aspects of rock music. The band sings mainly in Spanish, which sets them apart from many other stoner rock bands who mostly sing in English. Los Natas has released several albums and EPs throughout their career, including notable albums such as "Delmar" (1998), "Ciudad de Brahman" (1999), "Corsario Negro" (2002), and "El Hombre Montaña" (2006). Their music is characterized by heavy guitar riffs, pounding bass lines, and thunderous drumming, often accompanied by smooth and mellow vocals that add a layer of melodic complexity in contrast to the powerful instrumentals. In 2009, the band announced that they would be taking a hiatus, which resulted in several side projects for each of the member. It wasn't until 2011 that Los Natas officially disbanded, largely due to other projects and commitments undertaken by the band members. Over the years, Los Natas has been influential in the stoner rock and doom scene and has acquired a large fan base. Their records remain important staples within the collections of fans of the genre. Los Natas is remembered as one of the major pioneers of stoner rock and doom metal in the Spanish-speaking world, paving the way for other bands from the region to make a name for themselves in the international rock music scene.

Los Natas is an Argentine stoner rock band formed in Buenos Aires. The band consists of Sergio Chotsourian (guitar and vocals), Claudio Filadoro (bass), and Walter Broide (drums). Their music combines elements of stoner rock, doom metal, psychedelic rock, and heavy rock with hints of Argentine folk music. The band's name, "Los Natas," is derived from the word "satan" spelled backward ("natas" in Spanish). The band sings mainly in Spanish, which sets them apart from many other stoner rock bands who mostly sing in English. Los Natas has released several albums and EPs throughout their career, including notable albums such as "Delmar" (1998), "Ciudad de Brahman" (1999), "Corsario Negro" (2002), and "El Hombre Montaña" (2006). Their music is characterized by heavy guitar riffs, pounding bass lines, and thunderous drumming, often accompanied by smooth and mellow vocals that add a layer of melodic complexity in contrast to the powerful instrumentals. In 2009, the band announced that they would be taking a hiatus, which resulted in several side projects for each of the member. Los Natas officially disbanded in 2011, largely due to other projects and commitments undertaken by the band members. Over the years, Los Natas has been influential in the stoner rock and doom scene and has acquired a large fan base. Their records remain important staples within the collections of fans of the genre. Los Natas is remembered as one of the major pioneers of stoner rock and doom metal in the Spanish-speaking world, paving the way for other bands from the region to make a name for themselves in international rock music.

Figure 10: The original text and the revised text for the entity "Los Natas". gLM is GPT-4 and aLM is ChatGPT. The colors represent sentences revised to mitigate self-contradictions.

Naomi Watts is a British-Australian actress who was born on September 28, 1968, in Shoreham, Kent, England. She grew up in Australia and began her acting career on Australian television in the early 1990s. She later moved to the United States and gained worldwide recognition for her performances in films such as "Mulholland Drive" (2001), "21 Grams" (2003), "The Painted Veil" (2006), and "The Impossible" (2012). Naomi Watts has been praised for her versatility as an actress, as she has played a wide range of roles, from dramatic and intense characters to lighter and more comedic ones. She has received numerous awards and nominations for her work, including Academy Award and BAFTA nominations for Best Actress. In addition to her work on the big screen, Naomi Watts has also starred in several critically acclaimed television series, including "Twin Peaks" (2017), "The Loudest Voice" (2019), and "Birds of Prey" (2020). She is currently starring in the series "The Moon and the Sun" (2021). Naomi Watts is known for her private and low-key personal life. She was married to filmmaker Liev Schreiber from 2007 to 2021 and has two children with him. She is a committed humanitarian and is involved with several charitable organizations, including the David Lynch Foundation and Plan International.

Naomi Watts is a British-Australian actress who was born on September 28, 1968, in Shoreham, Kent, England. She grew up in Australia and began her acting career on Australian television in the early 1990s. She later moved to the United States and gained worldwide recognition for her performances in films such as "Mulholland Drive" (2001), and "21 Grams" (2003), "The Painted Veil" (2006), and "The Impossible" (2012). Naomi Watts has been praised for her versatility as an actress, as she has played a wide range of roles, from dramatic and intense characters to lighter and more comedic ones. She has received numerous awards and nominations for her work, including Academy Award and BAFTA nominations for Best Actress. Naomi Watts has starred in several critically acclaimed television series, including "Twin Peaks" (2017), "The Loudest Voice" (2019), and "Birds of Prey" (2020). Naomi Watts is known for her private and low-key personal life. She was married to filmmaker Liev Schreiber and has two children with him. She is a committed humanitarian and is involved with several charitable organizations, including the David Lynch Foundation and Plan International.

Figure 11: The original text and the revised text for the entity "Naomi Watts". gLM is Vicuna-13B and aLM is ChatGPT. Red color represents sentences removed from the original text, while other colors represent sentences revised to mitigate self-contradictions.

Table 13: A self-contradiction generated by Llama2-70B-Chat that cannot be verified using Wikipedia or text obtained by web search. ChatGPT detects this self-contradiction and decides to remove the sentence, which improves the overall factuality of the text description.

| | |
|---|---|
| [Prefix] | Prince Albert of Saxe-Altenburg (1843-1902) was a German prince and the last reigning duke of Saxe-Altenburg, a small duchy located in the eastern part of Germany. He was born in Altenburg, the capital of Saxe-Altenburg, and was the eldest son of Duke Ernest I and his first wife, Princess Marie of Hildburghausen. Prince Albert was educated at home and later studied law and politics at the University of Jena. He succeeded his father as duke in 1853, but due to his minority, a regency council ruled in his place until he came of age. In 1863, he was officially recognized as an adult and began ruling in his own right. During his reign, Prince Albert focused on modernizing the duchy and improving the lives of his subjects. He promoted the construction of railways, the expansion of agriculture and industry, and the development of education and healthcare. He also took an interest in art and culture, and was a patron of several artists and musicians. In 1866, Prince Albert played a significant role in the Austro-Prussian War, where he sided with the Austrian Empire against Prussia. However, his troops were defeated, and Saxe-Altenburg was subsequently annexed by Prussia. Prince Albert was forced to abdicate, and he was exiled to his estate in the Austrian Empire. Despite his exile, Prince Albert continued to be involved in politics and was a prominent figure in European society. |
| [Sentence pair] | He was known for his conservative views and his support for the Austrian monarchy. He was known for his liberal policies and his support for the arts. |
| [Mitigation] | Sentence removed. |

```
Prompt:
  I give you the beginning of a description about Diane Arkenstone.
  Then follow two statements.

  Description:
  Diane Arkenstone is an American musician, composer, and singer-songwriter.

  Statement 1:
  She was born on May 3, 1959, in Nebraska, USA.

  Statement 2:
  She was born in California.

  Please explain if the statements about Diane Arkenstone are contradictory.
  Provide your explanation only.

ChatGPT as aLM:
  The statements about Diane Arkenstone are contradictory. Statement 1 says she was born in
  Nebraska, while statement 2 says she was born in California. They cannot both be true at
  the same time.

Vicuna-13B as aLM:
  The statements about Diane Arkenstone are not contradictory. While statement 1 says she
  was born in Nebraska, statement 2 states that she was born in California. These two
  statements do not contradict each other because Nebraska and California are both states
  in the United States. It is possible that Diane Arkenstone was born in Nebraska but later
  moved to California. Alternatively, she may have been born in California but was simply
  mistaken about her place of birth when interviewed or quoted in the past. In either case,
  the two statements do not contradict each other.
```

Figure 12: An example where Vicuna-13B fails to correctly deduce contradictiveness. Vicuna-13B makes up three flawed reasons, highlighted by three different colors. In contrast, ChatGPT makes a succint and correct response.

```
System:
  You are a description generator. You are given the start of a description and a question that should be
  answered by the next sentence. You return the next sentence for the description.

Three-shot demonstrations using entities ``Douglas Adams'', ``Kayne West'', and ``Angela Merkel''.

Prompt:
  Here is the start of a description about William T. Freeman:
  William T. Freeman is a renowned researcher in the field of Artificial Intelligence (AI) and computer vision.

  Please generate the next sentence of this description.
  The generated sentence must fill the gap in this Subject;Predicate;Object triple: (He; was born; _)
  The sentence should contain as little other information as possible.

gLM: He was born in 1960.
```

Figure 13: Our full prompt for gLM.gen_sentence.

```
Prompt:
  I give you the beginning of a description about William T. Freeman.
  Then follow two statements.

  Description:
  William T. Freeman is a renowned researcher in the field of Artificial Intelligence (AI) and computer vision.

  Statement 1:
  He was born on August 15, 1955, in the United States.

  Statement 2:
  He was born in 1960.

  Please explain if the statements about William T. Freeman are contradictory. Provide your explanation only.

aLM:
  The statements about William T. Freeman are contradictory. Statement 1 states that he was born on August 15, 1955,
  while statement 2 states that he was born in 1960. These two statements cannot both be true at the same time.

Prompt: Please conclude whether the statements are contradictory with Yes or No.

aLM: Yes.
```

Figure 14: Our full prompt for `aLM.detect`.

```
Prompt:
  Here is the start of a description about William T. Freeman.
  William T. Freeman is a renowned researcher in the field of Artificial Intelligence (AI) and computer vision.

  Original Sentence:
  He was born on August 15, 1955, in the United States.

  This sentence originally followed the description. However, there is a contradiction with this sentence:
  He was born in 1960.

  Remove the conflicting information from the sentence, preserving as much valid information as possible.
  The result must fit well after the given description. Answer with the new sentence only.
  The new sentence must only contain information from the original sentence.

aLM: William T. Freeman was born in the United States.
```

Figure 15: Our full prompt for `aLM.revise`.

```
Prompt:
  I give you the beginning of a text answering the prompt "Please tell me about 2003 World
  Judo Championships - Men's 60 kg.".
  Then follow several statements. The first statement is the original statement of the text.
  The subsequent statements are additional evidence for you to verify the correctness of
  the original statement. If there is a contradiction between the original text and any
  of the additional evidences, you should conclude that the original statement is incorrect.

  Text:
  The 2003 World Judo Championships - Men's 60 kg was held in Osaka, Japan from September 12
  to September 14, 2003. The tournament saw the participation of 62 judokas from different
  countries.

  Original Statement:
  The gold medal in the men's 60 kg category was won by Japan's Tadahiro Nomura, who defeated
   South Korea's Choi Min-Ho in the final.

  Evidence 1:
  The gold medal in the men's 60 kg category was won by Yordanis Arencibia from Cuba.

  Evidence 2:
  The gold medal in the men's 60 kg was won by Hiroshi Izumi from Japan.

  Further 18 sampled statements, omitted for brevity

  Are there contradictions between the original statement and the provided evidence?
  Based on your reasoning about the above question and the evidences, what is your conclusion
  regarding the correctness of the original statement? Provide your explanation only.

aLM:
  There are contradictions between the original statement and the provided evidence.
  The original statement claims that Tadahiro Nomura from Japan won the gold medal in
  the men's 60 kg category, but the evidences provide different names as the winner,
  including Yordanis Arencibia from Cuba, Hiroshi Izumi from Japan, Ludwig Paischer
  from Austria, Ludovic Chammartin of Switzerland, and Muminjon Abdullaev of Uzbekistan.
  Therefore, the original statement is incorrect.
  Prompt: Please conclude whether the statement is incorrect a score between 0 (entirely
  incorrect) and 10 (fully correct). Answer just "Score: X" where X is your score

aLM: Score: 0
```

Figure 16: The prompt for detecting non-factuality in a sentence using 20 sampled alternatives using `gLM.gen_sentence`.

```
System:
  You are a description generator. You are given the start of a description. You return potential next sentences
  for the description.

Three-shot demonstrations using entities "Douglas Adams", "Kayne West", and "Angela Merkel".

Prompt:
  Here is the start of a description with the entity William T. Freeman:
  William T. Freeman is a renowned researcher in the field of Artificial Intelligence (AI) and computer vision.

  Please generate two valid continuations of this description.

gLM:
  1. His extensive research and contributions to the field have earned him numerous accolades, including induction
  to the National Academy of Engineering in 2021.

  2. Freeman's work has focused on developing algorithms for image and video processing, with a particular emphasis
  on creating machines that can understand and interpret visual data in the same way that humans do.
```

Figure 17: The full prompt of the baseline "Continue" for gLM.gen_sentence in Table 8.

```
System:
  You are a description generator. You are given the start of a description. You return the next sentence for
  the description.

Three-shot demonstrations using entities "Douglas Adams", "Kayne West", and "Angela Merkel".

Prompt:
  Here is the start of a description with the entity William T. Freeman:
  William T. Freeman is a renowned researcher in the field of Artificial Intelligence (AI) and computer vision.

  Please generate the next sentence of this description. It should be a rephrased version of this sentence:
  He was born on August 15, 1955, in the United States.

gLM:
  Born in the United States on August 15, 1955, William T. Freeman has become a distinguished figure
  in the realm of Artificial Intelligence (AI) and computer vision.
```

Figure 18: The full prompt of the baseline "Rephrase" for gLM.gen_sentence in Table 8.

```
Prompt:
  Here is the start of a description with the entity William T. Freeman:
  William T. Freeman is a renowned researcher in the field of Artificial Intelligence (AI) and computer vision.

  Please read the following sentence. Write at least two questions that can be answered by the information presented
  in the following sentence.
  Sentence:
  He was born on August 15, 1955, in the United States.

gLM:
  1. When was William T. Freeman born?

  2. Where was William T. Freeman born?

Prompt:
  I am going to ask you about William T. Freeman:
  William T. Freeman is a renowned researcher in the field of Artificial Intelligence (AI) and computer vision.

  Please answer the following question
  Sentence:
  When was William T. Freeman born?

gLM:
  I'm sorry, as an AI language model, I do not have access to real-time information. However, William T. Freeman
  was born on June 15, 1960, according to public records.
```

Figure 19: The full two-turn prompt of the baseline "Q&A" for gLM.gen_sentence in Table 8.

```
Prompt:
  I give you the beginning of a description about William T. Freeman. Then follow two statements.

  Description:
  William T. Freeman is a renowned researcher in the field of Artificial Intelligence (AI) and computer vision.

  Statement 1:
  He was born on August 15, 1955, in the United States.

  Statement 2:
  He was born in 1960.

  Are the two statements about William T. Freeman contradictory? Answer with either Yes or No.

aLM: Yes.
```

Figure 20: The full prompt of the baseline "Directly ask" for aLM.detect in Table 9.

```
Prompt:
  I give you the beginning of a description about William T. Freeman. Then follow two statements.

  Description:
  William T. Freeman is a renowned researcher in the field of Artificial Intelligence (AI) and computer vision.

  Statement 1:
  He was born on August 15, 1955, in the United States.

  Statement 2:
  He was born in 1960.

  Are the two statements about William T. Freeman contradictory? First, show your reasoning in a step-by-step fashion.
  Then conclude with yes or no.

aLM:
  Step 1: The description states that William T. Freeman is a renowned researcher in the field of AI and
  computer vision.
  Step 2: Statement 1 provides additional information about his birthdate, stating that he was born on August 15,
  1955, in the United States.
  Step 3: Statement 2 contradicts statement 1 by stating that he was born in 1960, which is a different year
  than the one provided in statement 1.
  Step 4: Therefore, the two statements about William T. Freeman are contradictory.
  Conclusion: Yes, the two statements about William T. Freeman are contradictory.
```

Figure 21: The full prompt of the baseline "Step-by-step" for aLM.detect in Table 9.

```
Prompt:
  I give you the beginning of a description about William T. Freeman. Then follow two statements.

  Description:
  William T. Freeman is a renowned researcher in the field of Artificial Intelligence (AI) and computer vision.

  Statement 1:
  He was born on August 15, 1955, in the United States.

  Statement 2:
  He was born in 1960.

  Please explain if the statements about William T. Freeman are contradictory. Provide your explanation only.

aLM (path 1):
  Yes, the statements about William T. Freeman are contradictory. Statement 1 gives the correct birthdate of August,
  15, 1955, while Statement 2 gives a different and incorrect birth year of 1960. Therefore, one statement is accurate
  while the other is incorrect.
aLM (path 2):
  Yes, the statements about William T. Freeman are contradictory. Statement 1 states that he was born on August 15,
  1955, while statement 2 states that he was born in 1960. These two statements cannot both be true at the same time.
aLM (path 3):
  Yes, the statements about William T. Freeman are contradictory. Statement 1 says he was born in 1955, while
  statement 2 says he was born in 1960. Both statements cannot be true at the same time, so they contradict each other.
aLM (path 4):
  Yes, the statements about William T. Freeman are contradictory. Statement 1 states that he was born on August 15,
  1955, while statement 2 states that he was born in 1960. Since the two statements provide different birth years,
  only one of them can be correct, making them contradictory.
aLM (path 5):
  Yes, the statements about William T. Freeman are contradictory. The first statement explicitly mentions
  his birthdate as August 15, 1955, whereas the second statement provides a different birth year altogether (1960).
  Since both statements cannot be simultaneously true, they are contradictory.

Prompt: Please conclude whether the statements are contradictory with Yes or No.

aLM (path 1, 2, and 4): Yes.
aLM (path 3 and 5): Yes, the statements are contradictory.
```

Figure 22: The full prompt of the baseline "Multi-path" for aLM.detect in Table 9.

