# OpenReview forum: "Self-contradictory Hallucinations of Large Language Models: Evaluation, Detection and Mitigation"
_ICLR.cc/2024/Conference — ICLR 2024 poster_

### Official Review · Reviewer_gQAk · 2023-10-31

**Soundness:** 3 good
**Presentation:** 2 fair
**Contribution:** 3 good
**Rating:** 6
**Confidence:** 4

**Summary:**

This work address a specific hallucination problem in large language models: self-contradition. More specifically, the authors proposed prompt-based approach, including generating, detecting and revising generated text. Extensive analysis shows prevalence of self-contradictions when LMs generate text for open-domain topics and proposed detection and mitigation method are shown to be effective.

**Strengths:**

- The motivations of the paper is important.
- This paper proposes a simple prompt-based method for triggering, detecting and migitating self-contradictions.
- The authors proposed a new eval human-annotated dataset.
-The authors provide code for reproducibility.
- Although authors demonstrate the proposed method in open-domain text generation task, same approach could be applied to other NLG tasks.
- The writing is clear

**Weaknesses:**

- Correct me if I'm wrong, the authors only evaluated aLM to detect output from gLM, how robust are aLMs to text generation tasks? that are not generated by "generating initial text, defining context and trigger" process?

- Last highlight in Revise, Section 5 is wrong.

**Questions:**

1. does the authors try other temperature in between? 0.5, 0.7 for both aLM and gLM?

2. please refer to 1st point in **Weakness** section.

---

> ### Author Response · Authors · 2023-11-16
> **Rebuttal**
>
> We would like to thank reviewer gQAk for providing insightful feedback. We hope that our rebuttal has addressed all of the reviewer’s questions, are happy to provide more details, and look forward to the reviewer’s response to our rebuttal.
>
> > ### Is aLM robust to another text generation task?
>
> Yes. We have added an evaluation on question answering, with the PopQA benchmark [1] suggested by reviewer a7zQ. The results show that our method can accurately detect a large number of self-contradictions, for both vanilla and retrieval-augmented generation. More details can be found in the global response.
>
> > ### Can you provide evaluation on different temperature values, for both aLM and gLM?
>
> Yes. We have provided this evaluation in the global response. The results show that across various temperatures (0, 0.25, 0.5, 0.75 and 1.0), self-contradictions are consistently prevalent and our method for detecting self-contradictions is robust.
>
> > ### Is the last highlight in paragraph Revise of Section 5 wrong?
>
> We have re-checked it and are confident that it is correct. According to https://wiki.factsider.com/william-t-freeman/, Freeman’s birth country is indeed the USA. If the reviewer still disagrees, we kindly request the reason.
>
> [1] When Not to Trust Language Models: Investigating Effectiveness of Parametric and Non-Parametric Memories. ACL 2023

---

> > ### Comment · Reviewer_gQAk · 2023-11-22
> >
> > Thanks for providing the rebuttal.
> >
> > The highlight section referred in the original comment is the green highlight section of "a faithful and fluent sentence". now after several thoughts, the highlight section is indeed correct. They were a bit confusing at first since the highlighted section is a description not math symbols.
> >
> > Thanks for the authors for providing additional experiments on temperature and on QA task. This solves my concerns.
> >
> > Although I acknowledge the remaining concerns of reviewer 3teD is valid, there is still values of current work, and i therefore increased the rating of the paper.

---

> > > ### Author Response · Authors · 2023-11-22
> > > **Thank you for the reply**
> > >
> > > We thank the reviewer for engaging in the discussion and are glad to hear that we could address their concerns.
> > >
> > > To avoid confusion, we have updated the paper to highlight only “sentence”.
> > >
> > > We have posted a new rebuttal to address the remaining concerns of reviewer 3teD. We welcome reviewer gQAk to read this rebuttal.

---

### Official Review · Reviewer_a7zQ · 2023-11-01

**Soundness:** 2 fair
**Presentation:** 3 good
**Contribution:** 3 good
**Rating:** 6
**Confidence:** 4

**Summary:**

This paper introduces a prompting-based method to detect and mitigate self-contradictory hallucinations. The proposed method generates texts based on a given prompt and subsequently evaluates these for self-contradiction by generating additional sentences. The text is then iteratively revised with the result of self-contradiction. Experimental results involving four LLMs, including GPT-4 and Llama-2-70B, suggest that the proposed approach effectively addresses self-contradictions.

**Strengths:**

The paper is clearly articulated and well-written. Moreover, it offers extensive experimentation with a range of LLMs, demonstrating a solid empirical foundation.

**Weaknesses:**

The primary issues with this paper include:

- The paper's central premise is that LLMs can produce self-contradictory hallucinations, and the proposed method aims to rectify using a prompting-based approach with LLMs. While it's acknowledged that LLMs can operate as zero-shot reasoners, their potential to induce hallucinations is also evident. Asserting that the authors correct LLM-induced hallucinations using LLMs (without external knowledge) appears paradoxical. This paper should incorporate a theoretical analysis explaining the efficacy of the proposed method in mitigating hallucinations.

- In the experiments, the verification of the text against Wikipedia is assessed, and various ratios are presented. However, a more in-depth exposition of the experimental setup should be shown. Specifically, manual checks were mentioned, but it's questionable whether the annotators thoroughly scanned the entirety of Wikipedia.

- While the paper claims that self-contradictions can be tackled without relying on external knowledge, it does utilize an "external" information extraction system to gather context.

- The term "topic" is employed in the paper, but Figure 4 indicates that experiments focused on specific entities. It seems that this paper is focusing on entities than on broader topics. If the text generated is primarily entity-centric, perhaps the experiments should have considered data from existing entity-centric QA research [1] even if the paper's emphasis on the open-domain text generation.

- The proposed iterative mitigation algorithm (Algorithm 3) requires an input n, which represents the number of mitigation iterations. The value of n seems to fluctuate based on the size of |x|. The paper should elucidate the chosen value for n and expound on how mitigation performance is influenced by this value.

[1] When Not to Trust Language Models: Investigating Effectiveness of Parametric and Non-Parametric Memories, ACL 2023

**Questions:**

Q1. How does the annotator manually verify contradictory information using Wikipedia?

---

> ### Author Response · Authors · 2023-11-16
> **Rebuttal**
>
> We would like to thank reviewer a7zQ for providing insightful feedback. We hope that our rebuttal has addressed all of the reviewer’s questions, are happy to provide more details, and look forward to the reviewer’s response to our rebuttal.
>
> > ### Why does your mitigation even work at all?
>
> Our mitigation approach **does not correct** hallucinations. Instead, it **removes** contradictory information that already exists in the two sentences. Our removal can be achieved by comparing the two sentences and does not rely on external knowledge. For instance, in Figure 3, the two sentences provide conflicting information about the birth date for William T. Freeman. This inconsistency can be removed without knowing Freeman’s actual birth date, by dropping the information about the birth date entirely from the sentence. It is rare but not impossible that LLMs introduce a new hallucination when generating the revised sentence. Since our mitigation algorithm is iterative, these new hallucinations are resolved in subsequent iterations.
>
> Our removal approach offers a novel perspective on mitigating hallucinations compared to the traditional correction approach. The motivation is to prevent LMs from generating mistakes when they are uncertain about specific facts. It aligns well with machine learning approaches that incorporate a reject or abstention option [2].
>
> > ### How do the annotators manually verify contradictory information using Wikipedia?
>
> The annotators read relevant Wikipedia articles and perform verification through a combination of human understanding and text searches. To enhance coverage, they are asked to examine various articles: (i) articles of all entities in the contradictory sentences; (ii) articles of relevant entities mentioned in (i); (iii) If non-English articles provided more information than English counterparts, the annotators used Google Translate to review non-English versions translated to English. To improve annotation accuracy, each piece of contradictory information must be examined by two annotators independently. Then, the two annotators discuss to resolve inconsistent cases and reach final decisions.
>
> We believe that the above protocol has resulted in high-quality verification annotations. We will incorporate this discussion into the paper.
>
> UPDATE: As suggested by reviewer 3teD, we additionally ask the annotators to use web search and examine broader online text. However, since Wikipedia already has a high coverage, the unverifiability numbers do not change significantly and our claims still hold.
>
> > ### Do you consider the information extraction system as external grounded knowledge?
>
> No. First, the information extraction system CompactIE does not retrieve any additional, external knowledge at any point. It only sees the sentences generated by gLM as input and outputs fact triples that are guaranteed to be spans of the input sentences. For example, from the sentence “X does Y”, it would extract the triple (“X”, “does”, “Y”). Second, CompactIE is based on a small BERT model with 110M parameters [3]. In contrast, external grounded knowledge typically refers to input-dependent text retrieved from considerably larger knowledge bases, such as Wikipedia. We will update the paper to clarify this.
>
> > ### For your work, “entity” seems a better term than “topic”.
>
> We agree and will update the paper to replace “topic” with “entity”.
>
> > ### Can you provide experiments on PopQA [1], an existing entity-centric QA dataset?
>
> Yes. We have provided this experiment in the global response. The results show that our method can accurately detect a large number of self-contradictions, for both vanilla and retrieval-augmented generation.
>
> > ### How is mitigation performance influenced by the value of n?
>
> The initial submission includes an ablation study on this. Please refer to the paragraph “Ablation Study on Mitigation” in Appendix B. Our ablation study shows that each iteration progressively removes self-contradictions, while preserving fluency and informativeness. The mitigation converges quickly after three iterations, which is why we choose $n=3$.
>
> > ### The value of n seems to fluctuate based on the size of |x|.
>
> We believe that there is a misunderstanding. The value of n is fixed at $n=3$ throughout all experiments presented in the paper and we do not intend to suggest at any point that n should be adapted based on |x|.
>
> [1] When Not to Trust Language Models: Investigating Effectiveness of Parametric and Non-Parametric Memories. ACL 2023
>
> [2] Machine Learning with a Reject Option: A survey. arXiv:2107.11277
>
> [3] CompactIE: compact facts in open information extraction. NAACL 2022

---

> > ### Author Response · Authors · 2023-11-22
> > **Rebuttal Reminder**
> >
> > We thank reviewer a7zQ again for the review comments, especially the suggestion of the PopQA benchmark. We believe to have addressed all concerns of the reviewer and look forward to the reviewer’s feedback. Your active engagement is invaluable for our work.

---

> > ### Comment · Reviewer_a7zQ · 2023-11-23
> >
> > Thank you for your response.
> >
> > Some of my concerns have been addressed, leading me to increase the score. However, I still have concern regarding the use of external knowledge in the information extraction (IE) system. Despite the size of the IE model, the proposed approach involves utilizing external knowledge. Therefore, it's not accurate to claim that these issues can be resolved without external knowledge.

---

### Official Review · Reviewer_3teD · 2023-11-07

**Soundness:** 3 good
**Presentation:** 3 good
**Contribution:** 2 fair
**Rating:** 6
**Confidence:** 4

**Summary:**

This paper proposes a pipeline to detect self-contradictory hallucinations generated by LLMs. Specifically designed prompts are used to ask LLMs to generate alternative answers to a question. Then an analysis LLM is asked to detect the potential contradiction between two answers and to revise them (to remove the contradiction).

The method is tested on a set of 30 topics and 360 descriptions generated by LLMs - a dataset created by the authors. The generated answers are verified manually. The method is compared with several baselines using other prompts. The results show that the proposed method can better detect contradictions than other baselines.

**Strengths:**

The paper examines an important problem of LLMs - hallucination. Despite the huge impact of the problem, there is a limited number of studies about the its detection. This paper offers an interesting solution to one type of hallucination.

The proposed method relies on existing LLMs. It is easy to implement and to replicate.

The authors provide a dataset and the tool that can be reused by other researchers.

The experimental results show some level of success with the proposed method, which outperforms several other pipelines (prompts).

Some ablation analysis is provided about the impact of different LLMs used in the generation and analysis steps.

**Weaknesses:**

The proposed method may detect one type of hallucination where alternative answers are contradictory. In reality, many hallucinations do not contain verifiable contradictions. The coverage of the method on different types of hallucination may be limited.
(edit after rebuttal: This is partially explained in the rebuttal)

The experiments are performed on a limited number of cases. Although a larger set of data is created, it cannot be used in a strict evaluation because of the lack of manual verification. I acknowledge the potential high cost for manual evaluation. However, a larger set of cases would make the experiments more meaningful. One possible solution is to leverage the existing datasets of hallucination. For example, a recent large dataset HalluEval (arXiv: 2305.11747) could be possibly used.
(edit after rebuttal: The number of topics used in the test is still limited. Therefore, the coverage is still questionable.)

The experimental results may suggest that there is a strong relationship between gLM and aLM. When ChatGPT is used as both, we can see that the performance is higher in most cases. This contradicts the intuition that GPT-4 is a more powerful LLM and should perform better than ChatGPT. Unfortunately, the paper does not report the case of GPT-4 as both gLM and aLM to see if the combination of GPT-4+GPT-4 is better than ChatGPT + ChatGPT. In any case, the fact that the combination of the same LLM for the two steps is better than when different LLMs are used for the steps may suggest that it would be better to use the same LLM for both steps. Some analysis is required to better understand the relationship between the two steps.
(edit after rebuttal: The answer in the rebuttal does not provide a plausible reason for this. It just restates the experiments.)

The revision tries to remove the contradictory part of the answers. This is a conservative revision. If one of the answer is correct, it would be better to keep it instead of removing it. From this perspective, it would be useful to refer to some external information source (e.g. Wikipedia). The authors argue that not doing so can make the approach independent of any external resource, but this may also make the method less reliable. Indeed, looking at the example of William T. Freeman, the generated birth dates are all wrong. A comparison with Wikipedia data would be able to tell that. An external resource would be valuable to the detection of hallucination.
(edit after rebuttal: Despite the explanation in the rebuttal, there is still not a strong motivation for the revision process.)

You mention that "a substantial portion of these self-contradictions (e.g., 35.8% for ChatGPT) cannot be verified on Wikipedia" to motivate the decision not to rely on external resources. However, you can verify on a much broader set of external resources. So the argument not to use an external resource is not very strong.
(edit after rebuttal: It is true that Wikipedia has a good coverage for general topics. It may miss many more recent or local events. Other web documents may be useful. Why not using web search to find possibly relevant information?)

The following description about informativeness is unclear (please revise): For informativeness, we calculate the ratio of non-contradictory sentence pairs (i.e., informative facts) in the revised text compared to the original text. Note that this ratio might exceed 100% because our mitigation can revise contradictory sentence pairs into non-contradictory ones.
(edit after rebuttal: the revised definition of informativeness is clearer.)

The paper proposes some reasonable prompts to generate answers and verify contradictions. The proposed prompts should be further motivated. The authors have not provided strong reasons to choose the specific prompts. The only way to see their superiority is through the comparisons in experiments with other alternative prompts.
(edit after rebuttal: The explanation about the prompts is reasonable. It still remains unclear why these specific prompts are chosen instead of other alternatives.)

The papers target long text generation in contrast to short answers targeted by some other papers. Why is the length of answers a key factor that distinguish this work from a set of existing studies? Would it still be possible to compare the proposed method with others on short answers?
(edit after rebuttal: If the method can be applied to both long and short texts, the paper should not emphasize the ability of the method for long texts. In fact, no specific means is taken to deal with long texts vs short texts.)

**Questions:**

Have you tested with different temperatures in decoding? This may produce different answers.

Can you compare the method with other methods that leverage external resources?

Would it be possible and meaningful to compare the proposed method with others on short answers?

---

> ### Author Response · Authors · 2023-11-16
> **Rebuttal Part 1**
>
> We would like to thank reviewer 3teD for providing insightful feedback. We hope that our rebuttal has addressed all of the reviewer’s questions, are happy to provide more details, and look forward to the reviewer’s response to our rebuttal.
>
> > ### Limitation and significance of self-contradiction.
>
> We agree with the reviewer that our work does not solve all types of hallucinations. In fact, we have explicitly stated this limitation in the initial submission (at the end of Section 7). However, we still believe our contribution is significant, because self-contradiction occurs frequently and our approach complements existing methods that retrieve external resources. More on the latter  is provided in subsequent answers and the global response.
>
> > ### Can you extend your evaluation to another existing dataset?
>
> Yes. We have added another evaluation on PopQA [1], a question-answering benchmark suggested by reviewer a7zQ. We have also performed human annotation for this experiment. The results show that our method can precisely detect a large number of self-contradictions, for both vanilla and retrieval-augmented generation. More details can be found in the global response and will be discussed in subsequent answers.
>
> > ### How comprehensive is your human-labeled evaluation dataset?
>
> While having added more evaluation data, we would like to emphasize that our current human-labeled dataset is already comprehensive. It covers 4 different LLMs, meaning that we perform separate annotations for each LLM. To address sampling variance, we generate and annotate 3 descriptions for each entity. Furthermore, to improve quality, each annotation label results from a joint effort of 2 annotators.
>
> > ### Can you provide more analysis on the relationship between aLM and gLM?
>
> Yes. The following table presents the detection results with all four combinations of ChatGPT and GPT-4 as gLM and aLM. We can observe (i) gLM and aLM should be different to achieve optimal F1 scores: ChatGPT-GTP-4 outperforms ChatGPT-ChatGPT, and GPT-4-ChatGPT outperforms GPT-4-GPT-4; (ii) Compared to ChatGPT, GPT-4 generates content that is more nuanced and more difficult to classify: in terms of F1 score, ChatGPT-ChatGPT outperforms GPT-4-ChatGPT, and ChatGPT-GPT-4 outperforms GPT-4-GPT-4.
>
> | gLM     | aLM     | P     | R     | F1    |
> |---------|---------|-------|-------|-------|
> | ChatGPT | ChatGPT | 84.2% | 83.2% | 83.7% |
> | ChatGPT | GPT-4   | 91.3% | 82.1% | 86.5% |
> | GPT-4   | ChatGPT | 80.1% | 79.7% | 79.9% |
> | GPT-4   | GPT-4   | 88.3% | 65.7% | 75.3% |
>
> > ### How does your conservative mitigation approach position in ML literature?
>
> Our approach for removing contradictory information offers a novel perspective on mitigating hallucinations compared to the traditional correction approach. The motivation is to prevent LMs from generating mistakes when they are uncertain about specific facts. It aligns well with machine learning approaches that incorporate a reject or abstention option [9].
>
> > ### If one of the answers is correct, it would be better to keep it instead of removing it.
>
> We acknowledge the value of keeping the original answers, and indeed, we do have such a feature in our tool at https://iclr9113.com/. Users can easily access the original answers by clicking on the revised answer.
>
> > ### What is your argument on the relationship between your method and other methods that retrieve external resources?
>
> The reviewer may have a misunderstanding of our paper, thinking that we advocate for an exclusive choice between our approach and methods that retrieve external resources. To clarify, our paper actually only argues that our approach **complements** retrieval-based methods. In other words, while these two approaches can be used alone, they can also co-exist and their combination can yield improved performance. For more details, please search for the word “complement” in our initial submission or examine the paragraph “Self-contradiction vs. Knowledge Retrieval”.
>
> We fully appreciate and agree with the reviewer on the value of retrieval-based methods. However, it is equally important to recognize their limitations, which have been extensively discussed [1, 2, 3, 4]. First, retrieval can be infeasible or prohibitively expensive in certain domains. Second, the retrieved information can be insufficient, inaccurate, or misleading, due to various factors such as the quality of the knowledge base or the retriever’s imprecision. As shown in our paper, even a resource as comprehensive as Wikipedia falls short in verifying a large portion of self-contradictions. Therefore, our approach serves as a valuable complementary solution to address these shortcomings.
>
> We have also strengthened the above argument experimentally with the PopQA benchmark [1], which can be found in the global response and covers self-contradiction detection on top of retrieval augmented model output.

---

> > ### Author Response · Authors · 2023-11-16
> > **Rebuttal Part 2**
> >
> > > ### Can you get significantly more verifiability by using more external resources than Wikipedia?
> >
> > We believe no. Wikipedia is already comprehensive for the purpose of our evaluation and is very likely to cover most facts in other resources. A study in Appendix A.5 of [5] supports this, revealing that “Wikipedia has a high coverage and mentions most of the important information that we were able to find from any other sources on the web”. Moreover, Wikipedia content is usually up-to-date and correct. Including other external resources may introduce uncertainties, potentially slowing down retrieval and increasing the risk of including irrelevant or wrong information. Treating Wikipedia as the general knowledge source is a common approach used by many prior works [5, 6, 7, 8]. We will add a discussion about this.
> >
> > > ### Can you revise your description about informativeness?
> >
> > Yes. We will revise the description to: “When a sentence does not induce contradiction, we consider it as informative. The evaluation of informativeness involves comparing the number of informative sentences in the original text with that in the revised text and calculating the ratio. Note that this ratio might exceed 100% because our mitigation can produce new informative sentences”. If the reviewer believes that the description still needs further improvement, we kindly request the reviewer to let us know.
> >
> > > ### How do you motivate your prompts?
> >
> > For “Trigger”, we have motivated that our prompt leverages cloze tests to enforce an appropriate level of constraint. For “Revise”, we have motivated that our prompt covers three aspects: removing contradiction, maintaining informativeness and coherency. For “Detect”, we will add a discussion on the benefit of our two-step chain-of-thought prompting approach.
> >
> > > ### Why is the length of answers a key factor that distinguishes this work from a set of existing studies?
> >
> > Addressing hallucinations in long text presents unique challenges. It requires handling dozens of facts, managing context, and maintaining coherency. Handling long text is explicitly listed as a future direction in the survey of [10]. This motivates our choice of open-domain text generation as our primary evaluation task. That being said, our approach is also applicable to shorter answers, as demonstrated with our experiments on PopQA [1] and the examples on our website at https://iclr9113.com/.
> >
> > > ### Can you compare with other methods on short answers? Can you compare with other methods that leverage external resources?
> >
> > Yes. Both questions are addressed by our experiments on PopQA [1], with results detailed in the global response. First, PopQA is a question-answering benchmark, where each question focuses on one specific fact answerable with short responses. Moreover, The PopQA paper [1] proposes an adaptive retrieval augmentation to improve factuality. We further demonstrate that our method accurately detects a significant number of self-contradictions on text generated with adaptive retrieval augmentation.
> >
> > > ### Can you provide evaluation on different temperature values, for both aLM and gLM?
> >
> > Yes. We have provided this evaluation in the global response. The results show that across various temperatures, self-contradictions are consistently prevalent and our method for addressing self-contradictions is robust.
> >
> > [1] When Not to Trust Language Models: Investigating Effectiveness of Parametric and Non-Parametric Memories. ACL 2023
> >
> > [2] Section 6. A Survey on Automated Fact-Checking. TACL 2022
> >
> > [3] Section 4.2.2 Survey on Factuality in Large Language Models: Knowledge, Retrieval and Domain-Specificity. arXiv:2310.07521
> >
> > [4] Section 6. A Survey on Retrieval-Augmented Text Generation. arXiv:2202.01110
> >
> > [5] FACTSCORE: Fine-grained Atomic Evaluation of Factual Precision in Long Form Text Generation. EMNLP 2023
> >
> > [6] Reading Wikipedia to Answer Open-Domain Questions. ACL 2017
> >
> > [7] Wizard of Wikipedia: Knowledge-powered Conversational Agents. ICLR 2019
> >
> > [8] KILT: a Benchmark for Knowledge Intensive Language Tasks. NAACL 2021
> >
> > [9] Machine Learning with a Reject Option: A survey. arXiv:2107.11277
> >
> > [10] Section 6.2. Survey of Hallucination in Natural Language Generation. ACM Computing Surveys Vol 55 Issue 12

---

> > > ### Comment · Reviewer_3teD · 2023-11-21
> > > **reaction to rebuttal**
> > >
> > > The rebuttal provides some explanation to the work. The additional experiments on temperature and on QA are good, showing that the method can be used in a different setting.
> > >
> > > Still, the explanation about the revision process, the lack of use of retrieval results and the justification of the specific prompts is not very convincing.
> > >
> > > It would also be useful to compare the method with others that deal with hallucinations for short texts. This should be possible, as the authors acknowledge that the method can be used for short texts. Then how does it compare to other existing methods?
> > >
> > > See the reactions in the review after rebuttal.
> > >
> > > Despite the remaining questions, I acknowledge the value of the additional tests and some useful explanations in the rebuttal. I therefore increased the rating of the paper.

---

> > > > ### Author Response · Authors · 2023-11-22
> > > > **Addressing Remaining Concerns**
> > > >
> > > > We thank the reviewer for engaging in the discussion and are glad that we could address some of their concerns. In this comment, we strive to address the remaining concerns.
> > > >
> > > > > ### Have you already considered using retrieval results in your evaluation?
> > > >
> > > > Yes. The PopQA paper [1] proposes an adaptive retrieval method to improve the factuality of QA. Our experiments on PopQA in the global response include a scenario where the results of adaptive retrieval are used to augment generation, as done in [1]. In this scenario, our method is still effective and finds a large number of self-contradictions (12.7% for ChatGPT and 21.5% for Llama2-70B-Chat). We find that self-contradictions occur specifically when the retrieval results do not contain relevant information.
> > > >
> > > > > ### Why is your removal-based revision valuable, even when retrieval-based methods exist?
> > > >
> > > > Retrieval-based methods come with limitations. First, retrieval can be infeasible or prohibitively expensive in certain domains. Second, the retrieved information can be insufficient, inaccurate, or misleading, due to various factors such as the quality of the knowledge base or the retriever’s imprecision. These limitations are extensively discussed in prior works [1, 2, 3, 4], explained at the end of Section 3 in our paper, and verified in our experiments. When ground truth information cannot be retrieved, the only way to improve factuality is to remove non-factual information, as done with our approach.
> > > >
> > > > > ### How does your approach compare to other methods on short text?
> > > >
> > > > Regarding experimental comparison, we have already evaluated our approach on top of PopQA generations. The results have demonstrated that our approach finds a significant number of self-contradictions on top of the adaptive retrieval method proposed by [1] to improve the factuality of short QA.
> > > >
> > > > Moreover, in the Table below, we provide a qualitative comparison between our approach and existing works that focus on short text. This comparison shows that our work is more comprehensive and has broader applicability.
> > > >
> > > > |  | consideration of long text | applicability to black-box LMs | independence from external knowledge | detection | mitigation |
> > > > |---|---|---|---|---|---|
> > > > | Our work | ✔ | ✔ | ✔ | ✔ | ✔ |
> > > > | [1] | ✘ | ✔ | ✘ | ✘ | ✔ |
> > > > | [5] | ✘ | ✘ | ✔ | ✔ | ✘ |
> > > > | [6] | ✘ | ✘ | ✔ | ✔ | ✘ |
> > > > | [7] | ✘ | ✘ | ✔ | ✔ | ✔ |
> > > > | [8] | ✘ | ✔ | ✔ | ✔ | ✘ |
> > > >
> > > > > ### Why do you highlight the ability of your approach for long text, if it is applicable to both long and short texts?
> > > >
> > > > We consider handling long text as one of our key contributions. First, it is a challenging task that requires handling dozens of facts, managing context, and maintaining coherency; Second, most prior work did not consider addressing long text (e.g., those in the table above), making our work novel.
> > > >
> > > > That being said, we have also included our results on short QA into the paper PDF and have provided sufficient discussion about the genericity of our work (e.g., the last paragraph of Section 1, the last paragraph of Section 5, the first paragraph of Section 6, Appendix C).
> > > >
> > > > > ### Addressing other concerns
> > > >
> > > > We discuss how we address other review concerns:
> > > > - **Improved coverage of entities**: We agree with the reviewer that the number of topics for our evaluation on open-domain text generation is not super large and scaling it up would cost significant human efforts. However, our evaluation in the global response has included 1500 topics from the PopQA dataset [1], which addresses the issue.
> > > > - **Using web search for annotating verifiability**: We have followed the review suggestion to additionally perform web search for annotating verifiability and have updated the numbers. As expected, the new numbers are not significantly different from previous ones and our claims still hold.
> > > > - **More reasoning for experiment results**: We have updated Appendix B to provide more analysis on our evaluation results. Specifically, we have provided the missing reasons for choosing the detection prompt, while we believe the motivation for choosing the trigger prompt has been sufficiently discussed. Moreover, we have provided plausible reasons on the experiment results regarding the relationship between aLM and gLM.
> > > >
> > > > [1] When Not to Trust Language Models: Investigating Effectiveness of Parametric and Non-Parametric Memories. ACL 2023
> > > >
> > > > [2] Section 2.2. A Survey on Automated Fact-Checking. TACL 2022
> > > >
> > > > [3] Section 5.3 Survey on Factuality in Large Language Models: Knowledge, Retrieval and Domain-Specificity. arXiv:2310.07521
> > > >
> > > > [4] Section 6. A Survey on Retrieval-Augmented Text Generation. arXiv:2202.01110
> > > >
> > > > [5] Semantic uncertainty: Linguistic invariances for uncertainty estimation in natural language generation. ICLR, 2023
> > > >
> > > > [6] The internal state of an LLM knows when it's lying. arXiv:2304.13734
> > > >
> > > > [7] Measuring and improving consistency in pretrained language models. TACL 2021
> > > >
> > > > [8] LM vs LM: detecting factual errors via cross-examination. arXiv:2305.13281

---

### Author Response · Authors · 2023-11-16
**Global Response**

We would like to thank all reviewers for the valuable feedback and insightful questions. We are happy to see positive reception on the importance of our motivation, the simplicity and effectiveness of our approach, the reusability of our tool and datasets, and clear writing. Before answering each review question in detail, we provide a significant extension of our experimental evaluation, with the goal of removing reviewers’ doubts on the generality and robustness of our work.

> ### Establish the effectiveness of your method on another text generation task (3teD, a7zQ, gQAk)

All three reviewers emphasized the importance of evaluating the generality of our approach. Therefore, we have performed an evaluation on question answering (QA) using the PopQA benchmark [1], as suggested by reviewer a7zQ. Adapting our approach to QA only requires a minor adjustment, i.e., modifying the first line of the prompts in Section 5 to include the question to be answered. **The evaluation results show that our approach can precisely detect a significant number of self-contradictions, for both vanilla and retrieval-augmented QA**. This reaffirms the universality of self-contradictions, the effectiveness of our method, and the claim that our approach complements retrieval-based methods.

In this evaluation, we use two gLMs (ChatGPT and Llama2-70B-Chat) and the same aLM (ChatGPT). Besides vanilla generation, we also experiment with augmentation with adaptive retrieval, as proposed in [1], a method known to improve factuality. We perform human annotation to obtain ground truth labels on whether the generated sentence pairs are self-contradictory. To make annotation feasible, we focus on 1.5K randomly sampled PopQA questions and annotate all sentence pairs that are predicted to be contradictory by our detection method. This enables us to compute detection precision. To improve annotation quality, the decision of each annotation label is a joint effort of two annotators.

The results are shown in the table below. For vanilla generation, our method predicts a significant ratio of self-contradictory answers (18.2% for ChatGPT and 38.0% for Llama2-70B-Chat), with high precision (83.2% for ChatGPT and 79.6% for Llama2-70B-Chat). This means that at least 18.6% (resp., 30.2%) of all answers generated by ChatGPT (resp., Llama2-70B-Chat) contain self-contradictions. When adaptive retrieval is enabled, the ratio of predicted self-contradictions decreases for both ChatGPT and Llama2-70B-Chat. However, our method still returns a large number of self-contradictions (12.7% and 21.5%) with high precision. A qualitative analysis shows that self-contradictions often arise when the retrieved knowledge lacks the information required for correctly answering the questions.

| gLM | retrieval? | self-contra. predicted | P |
|---|---|---|---|
| ChatGPT | No | 18.2% | 83.2% |
| ChatGPT | Yes | 12.7% | 83.8% |
| Llama2-70B-Chat | No | 38.0% | 79.6% |
| Llama2-70B-Chat | Yes | 21.5% | 74.2% |

> ### Show the consistency and robustness of your work across different temperature values (3teD, gQAk)

We have tested temperature values {0, 0.25, 0.5, 0.75, 1.0} for both gLM and aLM (ChatGPT in both cases for this experiment). **The results show that the claims of our paper hold across different temperature values**: (i) self-contradictions are consistently prevalent; (ii) our approach for addressing self-contradictions is robust.

We first fix the temperature of aLM to the default value 0 and vary the temperature of gLM in generating the initial text. For each temperature, gLM produces new text descriptions, which requires new expensive human annotations. To make human annotation feasible, we annotate all sentence pairs predicted by our detection method to be contradictory and report detection precision. Each annotation decision results from a joint effort of two annotators. The results are shown in the table below. We find that both the ratio of sentences predicted to be self-contradictory and the precision of our detection are consistent across different temperatures.

| temperature | self-contra. predicted | P |
|-------------|------------------------|-------|
| 0 | 17.4% | 84.1% |
| 0.25 | 17.8% | 81.3% |
| 0.5 | 17.2% | 79.2% |
| 0.75 | 16.6% | 83.3% |
| 1 | 18.3% | 83.4% |

Then, we fix the temperature of gLM to the default values and vary the temperature of aLM in detecting self-contradictions. We reuse existing human annotations in this experiment. The results are presented in the table below, showing that the precision, recall, and F1 scores have very small variance among different temperatures.

| temperature | P | R | F1 |
|-------------|-------|-------|-------|
| 0 | 84.3% | 83.8% | 84.0% |
| 0.25 | 83.4% | 84.4% | 83.9% |
| 0.5 | 83.3% | 83.8% | 83.6% |
| 0.75 | 80.9% | 82.7% | 81.8% |
| 1 | 84.2% | 83.2% | 83.7% |

[1] When Not to Trust Language Models: Investigating Effectiveness of Parametric and Non-Parametric Memories. ACL 2023

---

> ### Author Response · Authors · 2023-11-18
> **Integration of the rebuttal to the Paper PDF**
>
> We thank the reviewers again for their constructive feedback and eagerly anticipate their response to our rebuttal.
>
> We have performed additional evaluation of Llama2-70B-Chat on the PopQA benchmark and have updated the corresponding part of the Global Response. Moreover, we have incorporated the entire rebuttal into our paper and submitted a rebuttal revision of the PDF, making the following changes:
> - Our paper has seamlessly integrated our PopQA results, including a discussion on the instantiation of our approach to different text generation tasks such as question answering (Section 5) and the PopQA experiments (Section 6.2 and Appendix C). This integration has significantly improved the generality and practical usefulness of our work.
> - Appendix B now includes the other two suggested experiments, specifically different choices of temperature values and an analysis of the relationship between aLM and gLM. These additions demonstrate the robustness of our approach.
> - We have added or revised text to address all other review feedback, thereby enhancing the overall clarity, detail and coherence of the paper.
> - Several minor adjustments have been made to improve the writing quality.

---

### Author Response · Authors · 2023-11-21
**Rebuttal Reminder**

As the author-reviewer discussion period draws to a close, we kindly encourage the reviewers to acknowledge our rebuttal and pose any follow-up questions they might have. Your thoughtful engagement is highly appreciated.

---

### Meta-Review · Area_Chair_13fs · 2023-12-07

**Metareview:**

This paper investigates an important hallucination problem for LMs, namely self-contradiction. Self-contradictions means the LM generates two contradictory sentences within the same context. A prompt-based method is proposed to trigger, detect, and mitigate self-contradictions. The method is compared with several baselines using other prompts and the results show that the proposed method can better detect contradictions than the baselines.

The problem of self-contradiction in LMs is interesting and important. The proposed method is easy to implement and replicate. The dataset provided by the authors is a valuable resource.  However, the technical novelty of the proposed method is limited, as it relies on prompts over existing LLMs. The reviewers also pointed out a few issues about method and experiment design, and the authors addressed some of them during the rebuttal.

This paper is a borderline case. Considering the wide audience of the topic of self-contradiction, I am leaning towards weakly accept it.

**Justification For Why Not Higher Score:**

see the meta-review.

**Justification For Why Not Lower Score:**

see the meta-review.

---

### Decision · Program_Chairs · 2024-01-16

Accept (poster)